# Role of Intracellular and Extracellular Annexin A1 in MIA PaCa-2 Spheroids Formation and Drug Sensitivity

**DOI:** 10.3390/cancers14194764

**Published:** 2022-09-29

**Authors:** Nunzia Novizio, Raffaella Belvedere, Elva Morretta, Richard Tomasini, Maria Chiara Monti, Silvana Morello, Antonello Petrella

**Affiliations:** 1Department of Pharmacy, University of Salerno, Via Giovanni Paolo II 132, 84084 Fisciano, Italy; 2Cancer Research Center of Marseille, Institut Paoli-Calmettes, CNRS, UMR7258, Institut National de la Santé et de la Recherche Médicale (INSERM), U1068, University Aix-Marseille, 13009 Marseille, France

**Keywords:** annexin A1, pancreatic cancer, extracellular vesicles, spheroids, 3D models

## Abstract

**Simple Summary:**

In order to improve the investigation of pancreatic cancer (PC), often supported through analyzes two-dimensional (2D) cell monolayers, we proposed to create a spheroid-based in vitro three-dimensional (3D) model using wild-type (WT) and ANXA1 knock-out (KO) MIA PaCa-2 PC cells. However, the production of spheroids still represents a technical challenge. Here, we have developed a protocol to obtain well-organized spheroids and have proved that Annexin A1 (ANXA1) affects the spheroid formation, because the WT cells have a greater ability to form this 3D model when compared to the ANXA1 KO examples. We also investigated how ANXA1 action could influence the PC pharmacological response both in basal conditions and by mimicking a tumor system through the addition of autocrine EVs. ANXA1, via EVs, significantly improves the formation, the stability and the drug resistance of this model, particularly compared to the ANXA1 KO one, which shows a structural instability and a greater drug sensitivity.

**Abstract:**

Among solid tumors, pancreatic cancer (PC) remains a leading cause of death. In PC, the protein ANXA1 has been identified as an oncogenic factor acting in an autocrine/paracrine way, and also as a component of tumor-deriving extracellular vesicles. Here, we proposed the experimental protocol to obtain spheroids from the two cell lines, wild-type (WT) and Annexin A1 (ANXA1) knock-out (KO) MIA PaCa-2, this last previously obtained through CRISPR/Cas9 genome editing system. The use of three-dimensional (3D) models, like spheroids, can be useful to mimic tumor characteristics and for preclinical chemo-sensitivity studies. By using PC spheroids, we have assessed the activity of intracellular and extracellular ANXA1. Indeed, we have proved that the intracellular protein influences in vitro tumor development and growth by spheroids analysis, in addition to defining the modification about cell protein pattern in ANXA1 KO model compared to the WT one. Moreover, we have tested the response to FOLFIRINOX chemotherapy regimen whose cytostatic effect appeared notably increased in ANXA1 KO spheroids. Additionally, this study has highlighted that the extracellular ANXA1 action is strengthened through the EVs supporting spheroids growth and resistance to drug treatment, mainly affecting tumor progression. Thus, our data interestingly suggest the relevance of ANXA1 as a potential therapeutic PC marker.

## 1. Introduction

Pancreatic cancer (PC) correlates to poor prognosis and high mortality due to late diagnosis; in fact, it is the fourth worldwide cause of cancer deaths. At present, the incidence of PC is increasing year by year and the patients’ survival rate remains low because of remote metastasis and local recurrence, even after surgical treatment. PC is considered a morphologically and functionally heterogeneous tumor that can be difficult to treat [1,2]. Among the oncogenic protein factor identified for PC development, annexin A1 (ANXA1) has captured a great deal of attention over time. It belongs to the annexin family, known to bind the membrane phospholipids in a calcium-dependent manner [3,4,5]. Besides its anti-inflammatory activities, ANXA1 has been seen to play a role in malignant transformation, the activation of oncogenes, the inactivation of tumor suppressor genes, the induction of proliferation, apoptosis, cellular migration, invasion and metastasis [6,7,8,9]. Moreover, ANXA1 can act either as an anti-tumor or as a pro-tumor factor, depending on cellular localization, tumor type and stage, as well as on its expression levels. In this regard, high ANXA1 expression levels have been found in PC inducing the acquisition of an aggressive phenotype from both tumor and stromal cells [10,11,12,13,14,15,16,17,18,19]. Protein activity is strongly supported by its extracellular form, secreted by extracellular vesicles (EVs), such as exosomes [16,17,19,20]. This kind of EVs (40–100 nm diameter) is released from non-malignant and tumor cells, being crucial in cell-to-cell communication [21,22,22,23]. Their role has been described to be particularly important in the formation of pre-metastatic niches during tumor progression [24,25,26]. 

Usually, basic cancer research is performed by using conventional two-dimensional (2D) cell culture models. Nevertheless, this model fails to optimally represent the main tumor features and predict aggressiveness and drug sensitivity/resistance [27,28,29]. On the other hand, the establishment of in vivo systems is often linked to a number of ethical problems associated with the use of animal models [27]. Thus, in order to provide more physio-pathologically relevant in vitro models, researchers turned to three-dimensional (3D) cell culture systems, referred to as 3D models, to mimic in vivo tumor conditions by obtaining a more complex architecture, allowing more detailed and plausible analyses. Currently, the most widespread 3D model is represented by multicellular tumor spheroids (MCTS), made up of tumor cells or a co-culture of tumor and stromal cells, which acquire a spherical symmetry organized in a 3D arrangement. This latter is characterized by the stratification in: (i) an outer proliferative layer; (ii) a quiescent intermediate layer; (iii) a hypoxic and necrotic core, mainly due to the limited diffusion of oxygen and nutrients and the accumulation of catabolites and toxins [30]. This organization is considered the cause of a reduced permeability to drugs and the inhibition of apoptosis [31]. Many procedures have been developed to generate spheroids to improve the technical approach, included the reproducibility issues. In this study, we focused on the generation of a spheroid model composed by WT and ANXA1 KO MIA PaCa-2 cells in order to study how ANXA1 could affects the behavior of a PC 3D model. Particularly, we assessed the response to FOLFIRINOX treatment in the presence or not of the protein of our interest, ANXA1. The FOLFIRINOX chemotherapy regimen comprises of 5-fluorouracil, irinotecan and oxaliplatin, proving promising results in patients with advanced PC, administrated alone or in combination with gemcitabine and NAB-paclitaxel [32]. In separate clinical trials, patients treated for metastatic PC, FOLFIRINOX has shown a response rate of 31.6% compared to gemcitabine, with a response rate of 23%. Although clinical practice guidelines suggest that either regimen may be delivered as first-line therapy to patients with advanced PC, FOLFIRINOX has been favored leaving gemcitabine as an alternative for patients who do not tolerate FOLFIRINOX [33]. Numerous research teams have developed in vitro models to decipher PC resistance to gemcitabine, while others examined the individual effect of the three drugs included in the FOLFIRINOX protocol. These drugs, considered as individually entities, show different impacts on DNA stability by preventing synthesis, causing damage, or inhibiting repair [34,35,36].

Moreover, we also used Ac2-26 as the well-characterized mimetic peptide of the N-terminal region of ANXA1 [37,38], in order to examine the effects of this protein as exogenously administered to the established 3D system. 

## 2. Materials and Methods

### 2.1. Cell Culture 

WT MIA PaCa-2 and PANC-1 are immortalized epithelial cellular line of human pancreatic carcinoma and they were purchased from ATCC (ATCC CRL-1420 and ATCC CRL-1469, respectively, for MIA PaCa-2 and PANC-1 cells, Manassas, VA, USA). ANXA1 knock-out (KO) MIA PaCa-2 cells were obtained from the WT cells through the CRISPR/Cas9 genome editing system, as reported in [14], and kept in selection by 700 µg/mL neomycin (Euroclone; Milan, Italy). The cells were cultured as reported in [39] and maintained at 37 °C in a 5% CO_2_-95% air humidified atmosphere. 

### 2.2. Spheroids Generation and Area Analysis

The extracellular 3D cellular models were performed in 96 well U-bottom plates (#3799, Corning^®^ Costar^®^, New York, NY, USA). In this plate, 5000 tumor cells were seeded per well and were incubated for 10 days at 37 °C with 5% CO_2_, monitored in the Incucyte^®^ Live-Cell Analysis System (6 h repeat scanning, up to the end) until spheroid formation, and on day 5 the medium was replaced. After the spheroid development and growth, its size was reported in real time based on DF-Brightfield image analysis thanks to the Incucyte^®^ Spheroid Software Module.

### 2.3. Counting Live Cells in Spheroids

After 10 days, the cells in 3D models were first dissociated by phosphate buffered saline (PBS) 1x with 10% Trypsin and then they were counted by trypan blue exclusion using the Countess Automated Cell Counter (Invitrogen, Waltham, MA, USA).

### 2.4. Cell Viability for 3D Model

Cell viability was measured using CellTiterGlo (#G7571, Promega, Madison, Wisconsin, USA) on day 10, according to the supplier’s instructions. Briefly, an equal volume of CellTiterGlo reagent was added to the wells and incubated for 45 min at room temperature (RT) on a shaker. The cell suspension was then transferred to a black 96 well clear flat bottom plate and the relative luminescence units (RLU) were measured using a microplate reader (Synergy 2 Plate reader, Agilent BioTek, Santa Clara, CA, USA) [40,41,42].

### 2.5. Proteomic Analysis

WT and ANXA1 KO MIA PaCa-2 cells were suspended in 100 µL of a buffer comprising 8 M Urea, 0.5% *w*/*v* sodium deoxycholate (SDC) and 1× protease inhibitor cocktail (GeneSpin; Milan, Italy) in PBS. Cell suspensions were lysed through sonication (Vibra cell, SONICS; 1 min, 30% amplitude, 9.9 s pulses, on ice) and then submitted to centrifugation (21,000 rcf, 18 °C, 30 min, Centrifuge 5424 R, Eppendorf; Milan, Italy). Protein concentrations of the supernatants was determined through Bradford assay (Bio-Rad, Hercules, CA, USA). Then, equal protein amounts (20 µg) were treated with Laemmli buffer and heated at 95 °C for 5 min to run a 12% polyacrilamide short gel. Each lane was then cut in 3 pieces and submitted to an in situ tryptic digestion protocol as previously reported [43]. For the nano-flow RP-UPLC MS/MS analysis, peptides were solubilized in 30 µL of 10% Formic Acid (FA). Thus, 1 μL of each digest was analyzed on an Orbitrap Q-Exactive Classic Mass Spectrometer (ThermoFisher Scientific, Bremen, Germany) coupled to an UltiMate 3000 Ultra-High Pressure Liquid Chromatography (UPLC) system (ThermoFisher Scientific, Bremen), equipped with an EASY-Spray PepMAP^TM^ RSLC C18 column (3 μm, 100 Å, 75 μm × 50 cm, ThermoFisher Scientific, Bremen). Peptides elution was obtained at a flow rate of 300 nL/min with the following gradient: 1 min at 3% B, 1 min to 40 min to 28% B, 40 min to 41 min to 70% B (A: 95% H_2_O, 5% CH_3_CN, 0.1% AcOH; B: 95% CH_3_CN, 5% H_2_O, 0.1% AcOH). The mass spectrometer was operated in data-dependent acquisition mode. Full scan MS spectra were acquired as follows: scan range 375–1500 *m*/*z*, full-scan automatic gain control (AGC) target 3e6 at 70,000 resolution, maximum injection time 50 ms. MS2 spectra were generated for up to 8 precursors (normalized collision energy of 28%) and the fragment ions acquired at a resolution of 17,500 with an AGC target of 1e5 and a maximum injection time of 80 ms. Protein identification and *label-free* quantification were then achieved submitting the obtained raw files to Proteome Discoverer (version 2.4.1.15). *MSPepSearch* was employed to perform a spectral library search (NIST Human Orbitrap HCD Library, 1127970 spectra, September 2016) with a mass tolerance of 10 ppm for MS1 and 0.02 Da for MS2. The target False Discovery Rate (FDR) were set to 1% (strict) and 5% (relaxed). Then, MS/MS spectra were searched by *Sequest* against a reviewed *Homo sapiens* database (SwissProt, February 2022, 20,598 entries) with the following parameters: trypsin digestion; maximum of two missed cleavages; cysteine carbamidomethylation as fixed modification; methionine oxidization, protein N-terminal acetylation and/or demethylation as variable modifications. Mass tolerances and FDR were set as already reported. *Label-free* quantification was achieved exploiting both unique and razor peptides for proteins abundance calculation and a pairwise ratio-based approach was used to evaluate ANXA1 KO vs. WT MIA PaCa-2 proteins abundances. For each calculated ratio, a background-based *t*-test was performed. Proteins clustering was separately performed on both up- and down-regulated proteins through the web tool STRING using a high (0.9) confidence and hiding the disconnected nodes. Then, both up- and down-regulated proteins were annotated for their Gene Ontology (GO) and GO Biological Process terms (Appendix A), and then submitted to a GO and keywords enrichment analysis through the Database for Annotation, Visualization, and Integrated Discovery (DAVID). Thus, either the up- or the down-regulated proteins were used as gene lists, whereas all the identified proteins in both ANXA1 KO and WT MIA PaCa-2 samples served as a background proteome. The analysis was performed considering the up_kw_biological process in the Functional Annotations category and the GO_term_bp_all in the GO section with the default thresholds (i.e., count 2, EASE 0.1).

WT and ANXA1 KO MIA PaCa-2 cells were suspended in 100 µL of a buffer comprising 8 M Urea, 0.5% *w*/*v* sodium deoxycholate (SDC) and 1× protease inhibitor cocktail (GeneSpin; Milan, Italy) in PBS. Cell suspensions were lysed through sonication (Vibra cell, SONICS; 1 min, 30% amplitude, 9.9 s pulses, on ice) and then submitted to centrifugation (21,000 rcf, 18 °C, 30 min, Centrifuge 5424 R, Eppendorf; Milan, Italy). Protein concentrations of the supernatants was determined through Bradford assay (Bio-Rad, Hercules, CA, USA). Then, equal protein amounts (20 µg) were treated with Laemmli buffer and heated at 95 °C for 5 min to run a 12% polyacrilamide short gel. Each lane was then cut in 3 pieces and submitted to an in situ tryptic digestion protocol as previously reported [43]. For the nano-flow RP-UPLC MS/MS analysis, peptides were solubilized in 30 µL of 10% Formic Acid (FA). Thus, 1 μL of each digest was analyzed on an Orbitrap Q-Exactive Classic Mass Spectrometer (ThermoFisher Scientific, Bremen, Germany) coupled to an UltiMate 3000 Ultra-High Pressure Liquid Chromatography (UPLC) system (ThermoFisher Scientific, Bremen), equipped with an EASY-Spray PepMAP^TM^ RSLC C18 column (3 μm, 100 Å, 75 μm × 50 cm, ThermoFisher Scientific, Bremen). Peptides elution was obtained at a flow rate of 300 nL/min with the following gradient: 1 min at 3% B, 1 min to 40 min to 28% B, 40 min to 41 min to 70% B (A: 95% H_2_O, 5% CH_3_CN, 0.1% AcOH; B: 95% CH_3_CN, 5% H_2_O, 0.1% AcOH). The mass spectrometer was operated in data-dependent acquisition mode. Full scan MS spectra were acquired as follows: scan range 375–1500 *m*/*z*, full-scan automatic gain control (AGC) target 3e6 at 70,000 resolution, maximum injection time 50 ms. MS2 spectra were generated for up to 8 precursors (normalized collision energy of 28%) and the fragment ions acquired at a resolution of 17,500 with an AGC target of 1e5 and a maximum injection time of 80 ms. Protein identification and *label-free* quantification were then achieved submitting the obtained raw files to Proteome Discoverer (version 2.4.1.15). *MSPepSearch* was employed to perform a spectral library search (NIST Human Orbitrap HCD Library, 1127970 spectra, September 2016) with a mass tolerance of 10 ppm for MS1 and 0.02 Da for MS2. The target False Discovery Rate (FDR) were set to 1% (strict) and 5% (relaxed). Then, MS/MS spectra were searched by *Sequest* against a reviewed *Homo sapiens* database (SwissProt, February 2022, 20,598 entries) with the following parameters: trypsin digestion; maximum of two missed cleavages; cysteine carbamidomethylation as fixed modification; methionine oxidization, protein N-terminal acetylation and/or demethylation as variable modifications. Mass tolerances and FDR were set as already reported. *Label-free* quantification was achieved exploiting both unique and razor peptides for proteins abundance calculation and a pairwise ratio-based approach was used to evaluate ANXA1 KO vs. WT MIA PaCa-2 proteins abundances. For each calculated ratio, a background-based *t*-test was performed. Proteins clustering was separately performed on both up- and down-regulated proteins through the web tool STRING using a high (0.9) confidence and hiding the disconnected nodes. Then, both up- and down-regulated proteins were annotated for their Gene Ontology (GO) and GO Biological Process terms (Appendix A), and then submitted to a GO and keywords enrichment analysis through the Database for Annotation, Visualization, and Integrated Discovery (DAVID). Thus, either the up- or the down-regulated proteins were used as gene lists, whereas all the identified proteins in both ANXA1 KO and WT MIA PaCa-2 samples served as a background proteome. The analysis was performed considering the up_kw_biological process in the Functional Annotations category and the GO_term_bp_all in the GO section with the default thresholds (i.e., count 2, EASE 0.1).

### 2.6. Exosome Enrichment

The enrichment of exosomes (to which we generally refer to as extracellular vesicles, EVs) obtained from cell culture supernatants was performed as reported in [44]. Briefly, the seeded cells were washed with PBS and then incubated for 24 h in DMEM medium without FBS. Thus, the collected conditioned medium was centrifuged for 5 min at 300× *g* at RT to remove the detached cells, then for 10 min at 2000× *g* at 4 °C to remove dead cells. The recovered supernatant was centrifuged once more at 10,000× *g* for another 30 min at 4 °C to remove cellular debris. Finally, it was transferred into tubes and ultracentrifugated for 70 min at 100,000× g at 4 °C. Subsequently, the pellet was washed in PBS and re-ultracentrifuged at 100,000× *g* at 4 °C for 70 min. During the last step, the supernatant was removed and the pellet was re-suspended according to the experimental use. To validate the KO for ANXA1 in EVs we performed Western blotting, as reported in [16]. The fresh amount of exosomes administered to the cells was normalized at 20 µg of WT and ANXA1 KO MIA PaCa-2 EVs through the Bradford assay, as reported in [17]. The normalization allowed for the administration of the same amount of EVs to the cells, derived from the WT and ANXA1 KO MIA PaCa-2 cells, in all phases of the experiment. 

### 2.7. Spheroid Treatment

The preformed WT and ANXA1 KO spheroids on day 10 were treated with FOLFIRINOX (which we indicated as Folf in all figure panels) 25 µM (it means composed by 25 µM 5-fluorouracil, 11.25 µM irinotecan and 5.25 µM oxaliplatin) for 24 h and subsequently left in culture for 3 days with fresh medium and then analyzed. This procedure was adopted in order to see the effects of the drug treatment on the spheroid, which has a well-organized and compact structure that does not facilitate its entry. Concerning dual administration with EVs and Folf and Ac2-26 (the N-terminal mimetic ANXA1 peptide) and Folf, we performed a pre-treatment with EVs or Ac2-26 for 24 h, then Folf (24 h), changed the culture medium with fresh one, and finally the second treatment with EVs or Ac2-26, in the respective experimental points. In addition, in this case, we left the spheroids for 3 days in culture to be then analyzed.

### 2.8. Clonogenic Assay

WT and ANXA1 KO MIA PaCa-2 cells were seeded in 6-well plates and after the drug treatments were cultured until the 8th day in fresh medium. The clonogenic assay was performed as reported in [39]. Briefly, the cells were subsequently fixed with 4% p-formaldehyde, for 10 min, then with 100% methanol, for 20 min (both from Sigma-Aldrich; St. Louis, MO, USA) and finally stained with crystal violet at 0.5% *w*/*v* in a *v*/*v* solution of 20% methanol/distilled water (Merck Chemicals, Darmstadt, Germany) for 30 min at RT. After washing with deionized water, the colonies were photographed, and then, the dye was dissolved in 1% SDS and measured at 570 nm by spectrophotometer [33] (Titertek Multiskan MCC/340; Labsystems, Midland, ON, Canada), as confirmation of the result.

### 2.9. MTT Assay 

The WT and ANXA1 KO MIA PaCa-2 cells were treated and performed at the indicated times (24, 48 and 72 h) and cell viability was calculated as previously described in [45]. The optical density (OD) of each well was measured at 620 nm by spectrophotometer (Titertek Multiskan MCC/340; LabX; Midland, ON, Canada) [46].

### 2.10. Flow Cytometry for Cell Cycle and Cell Death 

WT and ANXA1 KO MIA PaCa-2 cells were harvested at a number of 1 × 10^5^ and analyzed as for cell cycle [47] as for cell death [48]. In both cases, after treatments cells were stained with propidium iodide (PI) (50 μg/mL; HiMedia Laboratories, Mumbai, India) and analyzed by FACScan cytometer (Becton-Dickinson, Franklin Lakes, NJ, USA). The percentage of apoptotic and necrotic cells were directly analyzed by the Cell Quest program, version 6.0, by which 1 × 10^5^ events have been evaluated by separating necrotic nuclei from apoptotic nuclei and, both of them, from viable cells in FSC/SSC and FL2/counts plots. For the assessment of cell cycle phases, viable cell samples were further analyzed through the ModFit LT analysis software, version 3.0. Gemcitabine (1 µM; Sigma-Aldrich; St. Louis, MO, USA) has been used as control in all experiments.

### 2.11. Western Blotting

Proteins extracted from WT and ANXA1 KO MIA PaCa-2 cells were examined by Sodium dodecyl sulphate-polyAcrylamide gel electrophoresis (SDS-PAGE). Protein content was extracted by freeze/thawing in lysis buffer containing protease inhibitors and estimated according to the Biorad protein assay (BIO-RAD, Hercules, CA, USA), as previously described [49]. A total of 40 µg of proteins were visualized using the chemioluminescence detection system (Amersham biosciences; Little Chalfont, UK) after incubation with primary antibodies against Cyclin E1 (rabbit polyclonal, 1:1000; Elabscience; Houston, TX, USA), Cyclin B1 (rabbit polyclonal, 1:1000; Elabscience; Houston, TX, USA), Cyclin A2 (rabbit polyclonal, 1:1000; Elabscience; Houston, TX, USA), GAPDH (mouse monoclonal 1:1000; Sigma-Aldrich; St. Louis, MO, USA). After incubation with the primary antibodies, the appropriate secondary antibody, either anti-mouse or anti-rabbit (diluted 1:5000; Sigma-Aldrich; St. Louis, MO, USA) was added for 1 h at room temperature. The blots were exposed to Las4000 (GE Healthcare Life Sciences, Little Chalfont, UK) and the relative band intensities, expressed by optical densitometry (OD), were determined using the ImageQuant software (GE Healthcare Life Sciences). 

### 2.12. Statistical Analysis 

Data analysis and statistical evaluations were carried out using Microsoft Excel. The number of independent experiments and *p*-values are indicated in the figure legends. All results are the mean ± standard deviation (SD) of at least three experiments performed in triplicate. Statistical comparisons between groups were made using one-way ANOVA and two-tailed *t*-test, as appropriated, followed by the Tukey’s multiple comparisons test. Differences were considered significant if *p* < 0.05 [50]. 

## 3. Results

### 3.1. The Ability of ANXA1 to Affect the Formation of MIA PaCa-2 Spheroids 

In this study, we have firstly improved the technical protocol to generate highly standardized spheroids with WT MIA PaCa-2 cells. PANC-1 cells were used as an experimental reference model (see Appendix A). The WT spheroids formation was compared with the ANXA1 KO examples. The establishment of this 3D model, referred to as WT and ANXA1 KO spheroids, respectively, has been reported in the Material and Methods section. In Figure 1A we showed that WT MIA PaCa-2 cells, after 10 days, macroscopically form a compact spheroid with an outer proliferative layer and a hypoxic and necrotic core, which can be correlated to active cellular interactions as well as compactness. On the other hand, the images of the ANXA1 KO MIA PaCa-2 cells organization show a structure comparable to an “aggregate”, meaning less tightly packed cells and not compact spherical cultures like spheroids (Figure 1A). These features, including a necrotic core, an inner layer of quiescent cells, and a layer of proliferating cells [51], have been assessed through the acquisition of 6 h repeat scanning images through the Incucyte^®^ Live-Cell Analysis System and analyzed in real time by the related software, as reported in the Material and Methods section. After 10 days, the formation of spheroids was also confirmed by the analysis of the number of cells that is greatly higher in WT spheroids, as well as the area and viability, than KO samples (Figure 1B). These analyses highlighted that mainly the WT spheroids are characterized by a marked proliferation, growth and cellular metabolic activity. This latter has been obtained as the generated ATP relative luminescent units (RLU) values and the increase of luminescence signal has been directly due to the increase of cell number and independently of area upon spheroid formation.

### 3.2. The Proteomic Analysis of WT and ANXA1 KO MIA PaCa-2 Spheroids

In order to characterize the WT and ANXA1 KO MIA PaCa-2 spheroids from a proteomic point of view, both samples were digested and analyzed by nano-LC-MSMS runs, twice. Following bioinformatic analyses carried out using the Proteome Discoverer software, around 3500 proteins were identified in both samples. Then, a label-free quantification has been carried out to disclose which proteins were up or down expressed in the ANXA1 KO samples versus WT samples. As reported in Appendix A, around 130 proteins were significantly over expressed (ratio ˃ 2-fold) and around 90 were significantly down expressed (ratio < 0.5-fold) in ANXA1 KO samples vs. WT. Figure 2A reports the volcano plot of significance versus fold change of all the quantified proteins, marking those showing a more pronounced expression variation in the two experimental conditions (Log_2_ of fold change >4 and <−4). As expected, Annexin A1 is highly present in the WT samples in respect to KO (ratio of 0.045, Log_2_ of fold change −4.47, *p*-value of 1.96 × 10^−5^,—Log_10_ *p*-value of 4.71). Furthermore, a STRING analysis was performed on all the differently regulated proteins: in particular some proteins more abundant in ANXA1 KO (ratio > 2) are involved in programmed cell death and apoptotic cleavage of cell adhesion proteins processes (Figure 2B, pathways in yellow, violet and cyano), whereas other proteins more abundant in WT (ratio < 0.5-folds) are involved in cell growth and proliferation (Figure 2C, pathways in red, green and blue), in accordance with the results reported in Figure 1B. Then, all the up- and down-regulated proteins were submitted to a GO (i.e., Gene Ontology) annotation for GO and GO Biological Processes terms, as reported in Appendix A. According to STRING, all of the up-regulated proteins related to apoptosis and cell death were highlighted in red, and in green all of the down-regulated protein terms related to mitosis and its spindle. Furthermore, an enrichment analysis has been performed through DAVID, with either the up- or down-regulated proteins as gene lists and all of the identified samples as background proteome. In more detail, proteins were enriched for the GO Biological Process and for the Keywords Biological Process terms and the obtained results are reported in Appendix A, respectively. As observable, for the down-regulated proteins both the Biological Process and Keywords analysis showed a major enrichment for mitosis-related terms. For the up-regulated proteins, the most enriched terms for the Biological Process are reported in Appendix A. Nevertheless, they also comprise terms related to cell-killing, accordingly with STRIBNG results and, for the Keywords, to cell adhesion (Appendix A).

### 3.3. The Effect of Chemotherapy Drugs on 2D Monolayers of MIA PaCa-2 PC Cells

The conventional 2D cell culture monolayers are usually used as a model for preliminary investigations of therapeutic drug sensitivity or resistance. Here, we have performed a MTT assay on WT and ANXA1 KO MIA PaCa-2 cells to test the action of FOLFIRINOX. Figure 3A showed greater sensitivity of ANXA1 KO MIA PaCa-2 cells to FOLFIRINOX (Folf in figures) (25 µM) at both 24 and 48 h of treatment compared to WT cells, as evident by the decrease of cell proliferation, and both compared to gemcitabine (1 µM) we chose as reference. The effects of FOLFIRINOX became similar after 72 h. This finding was also confirmed by the clonogenic test. The ability to growth forming colonies resulted inhibited by FOLFIRINOX, particularly in ANXA1 KO cells (Figure 3B). This result can be observed both in the brightfield images and in the histograms relating to the dissolution by SDS of the crystal violet adsorbed by the cells (Figure 3B). Moreover, to further study the sensitivity to FOLFIRINOX, we evaluated the apoptosis, necrosis and cell cycle on monolayers of PC cells. As shown in Figure 3C,D, the treatment with FOLFIRINOX increased the cell death, as assessed by the number of cells undergoing apoptosis and necrosis, particularly for the ANXA1 KO cell line compared to the WT counterpart. The necrotic effect on ANXA1 KO PC line further became more evident at 48 and 72 h of drug treatment. Finally, the analysis of cell cycle reinforced the evaluation of the fair sensitivity of WT cells in presence of FOLFIRINOX (Figure 3E), with a decrease in the G1 phase and an increase in the S one at 48 and 72 h of treatment. On the other hand, the ANXA1 KO MIA PaCa-2 cells, more sensitive to FOLFIRINOX, showed a cell cycle arrest, evidenced by a strong decrease of cells in G1 phase and a following increase in S and G2/M samples, compared to both the untreated control and to the WT examples. 

### 3.4. The Effect of Chemotherapy Drugs on WT and ANXA1 KO Spheroids 

Once the WT and ANXA1 KO spheroids were established and characterized, we treated them with FOLFIRINOX as reported in the Material and Methods section. The analysis of 3D model revealed significant differences between WT and ANXA1 KO spheroids in sensitivity to FOLFIRINOX treatment (Figure 4A), as observed in monolayer cultures. In particular, WT spheroids continued to stay in a huge aggregate when they were treated with FOLFIRINOX, although the cell number and area resulted decreased if compared to the control (Figure 4B,C). On the other hand, ANXA1 KO spheroids, characterized by a minor compactness compared to the WT counterpart as described above (see Figure 1), after treatment with FOLFIRINOX, showed a marked reduction in the number of cells. Additionally, the ANXA1 KO spheroids size appeared decreased, even if in a less evident way compared to the FOLFIRINOX-treated WT spheroids. Nevertheless, the main feature we found in this case has been the appearance of a notable amount of empty spaces due to the loss of compactness, likely derived from the loss of cell-cell interactions (bright field images reported in Figure 4A) as one of the results of the less aggressive phenotype shown by PC cells without ANXA1. Different from the monolayer model, which has been treated following a time-dependent curve, for the spheroids a single experimental time has been chosen (24 h) depending of their physiological state, based on the spheroid size, the individual and cell type–specific behavior of the tumor cells, the cell density within the spheroid and also directly or indirectly on the culture time. Thus, all these conditions reach the best feature in less than 48 h, a time range during which spheroid also has to be tested after treatment [52]. In a second instance, spheroids have been left for 3 days without drugs after the single FOLFIRINOX administration to unravel the delayed effect of FOLFIRINOX on cancer cells, since it has been shown that highest cell mortality occurs several days after treatment arrest [53].

Figure 4D shows that FOLFIRINOX induced a significant increase in spheroids apoptosis and necrosis, particularly for the ANXA1 KO samples, whose higher percentage of death as a proper characteristic has been confirmed with respect to the WT related control. Thus, in this case, in the presence or in absence of ANXA1, the spheroids apoptosis and necrosis followed the same trend in response to the FOLFIRINOX. Finally, as shown in Figure 4E, both models of spheroids have undergone a notable alteration in cell cycle with the decrease of G1 phase and a following increase of S and G2/M examples. However, even in this case the ANXA1 KO examples have been the most sensitive spheroids. This data has been further confirmed by the Western blot analysis of cyclins. Thus, in Figure 4F it is shown the increase of the cyclins A2 and B1 after FOLFIRINOX treatment both in WT and ANXA1 KO spheroids referring to the induction of the transition to the G2/M phases. These differences appeared more significant for the ANXA1 KO samples (treated vs. not treated). It is to note that the not treated ANXA1 KO points expressed a very low level of both these proteins highlighting the poorer ability of these cells to proliferate in the 3D organization. On the other hand, the cyclin E1, as indicator of G1 phase, notably decreased in accordance to the enter G2/M phase induced by the FOLFIRINOX treatment. Additionally, in this case, the differences among treated and not treated spheroids appeared more evident for ANXA1 KO samples. The densitometry analysis of Western blotting is reported in Appendix A.

### 3.5. The Autocrine EVs Effect on Chemotherapy on MIA PaCa-2 Spheroids 

In order to investigate the role of ANXA1 as content of EVs in the response to drug treatment, we administered EVs before and after the FOLFIRINOX, as reported in the Material and Methods section. These EVs are obtained from cell culture supernatants of WT and ANXA1 KO MIA PaCa-2 seeded in monolayers, as we had already performed in [16]. These microvesicles are usually characterized through Field Emission Scanning Electron Microscopy (FE-SEM) and by dynamic light-scattering (DLS) analyses highlighting rounded particles ranging from 30 to 180 nm in diameter. In addition, through Western blot analysis, it was confirmed that the presence of the proteins TSG101, CD63 and CD81, as positive control, together with the absence of the calreticulin, as a negative one. Finally, the lack of ANXA1 in ANXA1 KO MIA PaCa-2 deriving EVs was tested [16,17]. Here, we focused on the spheroid growth and morphology first by bright field images taken at the various experimental points pre (on day 10 after cells seeding) and post FOLFIRINOX-treatment (Figure 5A,B, for WT and ANXA1 KO spheroids, respectively). In detail, we found that WT spheroids continued to keep a solid and compact structure when they are treated with autocrine EVs (Figure 5A(c,c’)), containing ANXA1. This outcome was confirmed by cell count and area analysis; in fact, in the presence of EVs, the spheroids appeared larger in size (Figure 5E) meaning a higher proliferation rate as also assessed through cell number count (Figure 5C). On the contrary, the cell mass appeared scattered in the presence of FOLFIRINOX, both in spheroids pre-treated with EVs (Figure 5A(e,e’)), and in samples receiving EVs in pre- and post-treatment of FOLFIRINOX (Figure 5A(f,f’)). Moreover, the WT spheroids pre-treated with EVs and receiving FOLFIRINOX reached a weakly smaller size (as represented through area, Figure 5E, and cell number, Figure 5C) in comparison to the experimental point with only EVs. This outcome proves that in the presence of autocrine EVs, added after FOLFIRINOX, WT spheroids lose their fair drug sensitivity. The Ac2-26 administration confirmed that the exogenous ANXA1 is able to induce very similar effects of WT EVs (Figure 5A(b,b’,g,g’,h,h’)).

Then, we analyzed under the same conditions, the behavior of the ANXA1 KO spheroids (Figure 5B). Therefore, with the autocrine ANXA1 KO EVs (Figure 5B(k,k’)), referred in Figure 5 as KO EVs, the spheroids growth slightly improved as indicated by the cell number (Figure 5D), and by compactness (see representative bright field images in Figure 5B). The area (Figure 5F), evaluated in the presence of one administration of EVs (pre-treatment) and FOLFIRINOX (Figure 5B(m,m’)), was found weakly more compact than the point with the only FOLFIRINOX (Figure 5B(l-l’)), that appeared crumbling, with empty areas. While with the second additional EVs administration, the KO spheroids (Figure 5B(n,n’)) appeared more thickened and they achieved very similar morphological characteristics to the experimental point with only autocrine KO EVs (Figure 5B(k,k’)). Furthermore, the KO spheroids in the presence of Ac2-26 alone (images Figure 5B(j,j’)), and Ac2-26 as pre- (images Figure 5B(g,g’)) or post-treatment to FOLFIRINOX (Figure 5B(h,h’)), have significantly refined the behavior, in term of compactness and increased number of cells (Figure 5D–F). This result acquires a greater importance if compared to the related untreated control (Figure 5B(i,i’)) and the experimental point with FOLFIRINOX alone (Figure 5B(l,l’)), and finally in respect to the spheroids with autocrine EVs (Figure 5B(c,c’,e,e’,f,f’)), therefore without ANXA1.

### 3.6. The Autocrine EVs Effects on Chemotherapy on WT and ANXA1 KO Spheroids 

Finally, we evaluated how the WT and ANXA1 KO EVs could influence the death and the cycle of the cell components in spheroids, if administered alone or in combination with Folf (as for the same experimental procedures described above). As shown in Figure 6, apoptosis and necrosis (Figure 6A) of the WT spheroids increased after the EVs administration both alone and particularly in the EVs pre-treated spheroids with Folf. However, when EVs were added after Folf, cell death notably decreased. This finding was also confirmed in the presence of Ac2-26 and Folf together, but the administration of Ac2-26 alone did not induce any change in cell death compared to the untreated control. On the other hand, the KO spheroids, which already showed an intense basal necrosis in the control, showed a further increase of this process in presence of Folf both alone and in co-treatment with EVs. Then, apoptosis strongly increased with Folf in the case of EVs pretreatment, while it significantly decreased with EVs post-treatment. When Ac2-26 was added as pre-treatment and then as post-treatment to Folf, cell apoptosis decreased as well as necrosis (Figure 6B). Finally, once examined in the manner described above, we found that the WT (Figure 6C) and ANXA1 KO (Figure 6D) spheroids have undergone an alteration of cell cycle when treated with Folf, either alone or in co-administration with EVs or Ac2-26. A decrease of the number of cells in phase G1 was evident for WT spheroid model, with a related increase of cells distributed in S and G2/M phases in all the experimental points with Folf, in particular with the EVs and Ac2-26, which alone has promoted the increase of G1 phase. Finally, the ANXA1 KO spheroids followed the same behavior of the WT counterpart but, also in this case, they were more sensitive to treatment with Folf both alone and in addition to EVs or Ac2-26.

## 4. Discussion

2D in vitro models are widely used for the study of cellular physiology and tumorigenesis [54]. Nevertheless, they retain many limits about specific investigation, as for the evaluation of response to pharmacological treatments [27,28,29]. These limiting features have been overcome by the establishment of in vivo systems which, in turn, possess important ethic and technical obstacles. In this scenario, it has become necessary for the development of 3D models reflecting a link between the in vitro and the in vivo examples [27,28]. Indeed, 3D cell models, like spheroids, have been described as able to accurately reproduce the tumor and to provide critical insight on therapy resistance [29]. However, the production of 3D tumor models still represents a challenge as not all tumor cell lines form morphologically regular spheroids avoiding disaggregation in case of manipulation. In PC, the MIA PaCa-2 cells represent an in vitro line usually utilized for functional and drug sensitivity studies, but they are known to hardly be disposed to form stable spheroids useful for subsequent analyses [55]. In our previous work, we used the hanging drop method to produce MIA PaCa-2 spheroids [39]. However, this method proved to be not easily manageable to allow all kind of analyses. For this reason, here, we perfected the U-bottom plate method by defining the best conditions to overpass a great part of the technical limitations. Thus, we have obtained well-organized spheroids, as confirmed by the morphological and morphometric analyses, which we have then used as a model for our study. First of all, we could prove that the presence of ANXA1 in the WT spheroids is decisive for the creation of this tumor model reflecting the main PC features. In fact, the ANXA1 KO cells do not form very compact spheroids with proliferating margins, and a dense and necrotic core, which can be generally expected by spheroids. This lack of compactness, assimilable to the general characteristics of cell aggregates more than specific 3D models, could be correlated to the absence of the protein of our interest, which has been shown in-depth to be involved in the maintenance of a pro-motility cytoskeletal organization and of a more aggressive tumor phenotype [14]. Indeed, only the WT counterpart has acquired tumor-like characteristics as proved also by the proteomic profile. By this evaluation, several proteins involved in cell growth and proliferation have been found overexpressed in WT spheroids, while other examples important for cell death and apoptotic cleavage of cell adhesion proteins have been highlighted in ANXA1 KO counterpart. Furthermore, among the other proteins revealed as disregulated, in WT spheroids we have interestingly observed the overexpression of proteins involved in cell migration/invasion and PC progression, such as tetraspanin 1 (TSPAN1), in the acquisition of a more aggressive phenotype such as transglutaminase 1 (TGM1) and transglutaminase 3 (TGM3) proteins [56,57]. In addition, the higher expression of fibroblast growth factor 2 (FGF2) and CD14 can be associated to the great ability of WT spheroids to influence the other cellular components of tumor microenvironment, such as fibroblasts and monocytes positively supporting tumor development [58,59]. Since these proteins have been not significantly found in the spheroids without ANXA1, this kind of model is of marked importance and interest, and a deepened and detailed assessment of all protein modification should be considered notably useful to better understand the molecular aspect deriving from the ANXA1 activity. 

Before proceeding with the investigation about spheroids drug sensitivity, we have analyzed the response of the cells in 2D monolayer to FOLFIRINOX, using Gemcitabine as reference. We choose this mix of drugs because it is the most used chemotherapy approach in patients with advanced PC, both alone and in combination with Gemcitabine and other adjuvants. Thus, the finding of a higher sensitivity of MIA PaCa-2 cells without ANXA1 already provided an interesting starting point to our tests on 3D related models. Actually, the ANXA1 KO spheroids also confirmed this appealing outcome showing themselves as morphologically smaller and disaggregated, with empty areas in the core. In detail, drug treatment strongly induced a significant alteration of the cell cycle, with a decrease of the number of cells in the G1 phase and a correspondent increase of S and G2/M samples. Moreover, this data has been corroborated by cyclins profile. In fact, the enhancing levels of cyclins A2 and B1, above all in ANXA1 KO spheroids, demonstrated the more marked S and G2/M phases alterations induced by FOLFIRINOX [60,61,62]. On the contrary, cyclin E1 appeared strongly decreased in FOLFIRINOX-treated WT spheroids until it almost disappeared in treated KO examples, meaning the stop of the start transition cell cycle [63]. In the contest of the definition of a clinical protocol for FOLFIRINOX use, these outcomes acquire a great interest since they are in absolute tendency with what was demonstrated about the significant accumulation of cells organized in 3D models in the S and G2/M phases of the cycle, a response mainly due to 5-FU [53]. Another important cue we have to underline is the similar trend about the response to FOLFIRINOX of WT and ANXA1 KO MIA PaCa-2 cells distributed in monolayer and in spheroids. The absence of ANXA1 represents a fundamental parameter for the acquisition of a higher sensitivity to this chemotherapy approach. Indeed, there are also experimental evidences demonstrating that several drugs may exclusively be effective in 3D but not 2D cultures, as it has been seen in target-specific treatment modalities for which the molecular target is expressed only or particularly in a 3D environment [52,64,65,66]. According to these concerns, in our case it has been necessary to test drug efficacy before by classical monolayer test assays, and then on more complex spheroid structures. 

Taken together, this information appears able to add an important step in knowledge regarding the intracellular role of ANXA1 in PC drug response, even if no specific evidence for a possible direct correlation between ANXA1 and FOLFIRINOX still exists in PC. Additionally, the extracellular form of ANXA1 has been studied in-depth, proving its crucial role as a component of EVs in PC development [4,7,11,14,16,17,19]. Here, taking advantage of the 3D model we have generated, we investigated the FOLFIRINOX effects linked to the presence or not of ANXA1 by using autocrine EVs [16,17,19]. In this way, we could stabilize a model which likely recreates the tumor behavior inserted in its own microenvironment. This specific action is also supplied by the autocrine loop EVs/tumor cells, from which EVs derive, that generally preserves tumor survival, growth and drug resistance. Thus, we have proven that the WT EVs favor the growth and the size of the spheroid compared to the ANXA1 KO counterpart. It is important to underline that these findings have been further confirmed by the exogenous administration of Ac2-26 which, as the N-terminal mimetic ANXA1 peptide, is able to carry out the biological and physiopathological functions of the extracellular form of the protein of our interest [7,9,14]. In fact, also in the case of Ac2-26 treatment, ANXA1 KO spheroids grow up with a related partial reversion of the response to FOLFIRINOX. In this way, we have assisted in the acquisition of a very similar behavior of ANXA1 KO spheroids, compared to the WT samples, in term of drug resistance, strongly reverting the higher basal sensitivity we have observed without the exogenous ANXA1 administration. These findings reinforce our hypothesis for which, also on spheroids, the oncogenic effects of ANXA1 could be mediated by the extracellular form of this protein, as the content of the membrane EVs protein pattern, to trigger cell activation. In particular, its pro-tumor mechanism of action can be due to the ability to bind the formyl peptide receptors (FPRs), in particular the isoforms FPR1 and FPR2 [14,17,39]. Thus, it is possible to suggest that this protein can trigger the cascade of reaction downstream of the activation of these receptors on the PC spheroids. On the other hand, it was interesting to examine the resistance of WT spheroids to the FOLFIRINOX, which is robustly reinforced in the presence of the related WT EVs. This finding supports the establishment of the already mentioned positive loop for which the EVs, in an autocrine manner, notably encourage tumor progression not only in the promotion of metastatization, but also as expedient to reduce sensitivity to any chemotherapy approach.

## 5. Conclusions

This work allowed us to pave the way in the establishment of PC uniform spheroids, which can be used for real-time monitoring of tumor drug response. We based our study on the fact that PC is often associated with therapeutic resistance, and the ANXA1 affects PC progression, like an oncogenic factor, in autocrine and/or paracrine manner, and also via EVs [11,16,17,19]. Thus, we used WT and ANXA1 KO models to investigate the role of this protein in the generation and growth of spheroids and their pharmacological response, both in basal conditions and by mimicking a tumor system through the addition of autocrine EVs. In this regard, we proved that ANXA1 positively affects spheroid formation, as justified by the behavior of the WT cells, which show a greater ability of compacting in this 3D model compared to ANXA1 KO samples. Additionally, the extracellular form of ANXA1, also as content of EVs, is capable of reinforcing resistance to FOLFIRINOX. Thus, targeting the interaction between ANXA1/EVs might be a potential therapeutic strategy for PC. Further investigations are necessary to deepen the examination of WT and ANXA1 KO spheroids in order to better define the main differences, directly or indirectly linked to the protein of our interest. Moreover, it will be particularly interesting to amplify the assessment on the impact of EVs with and/or without ANXA1 on co-culture spheroids generated in association with other cell populations surrounding the primary tumor, such as stromal cells. Finally, it would be intriguing to perform in vivo studies to explore the answer of this complex to the FOLFIRINOX by implanting WT and ANXA1 KO spheroids in mice pancreas.

## Figures and Tables

**Figure 1 cancers-14-04764-f001:**
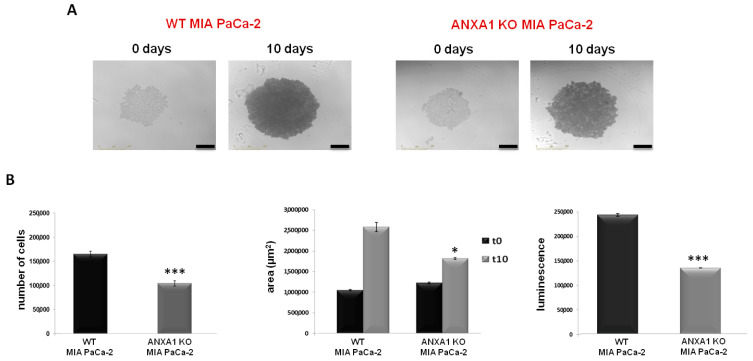
WT and ANXA1 KO MIA PaCa-2 spheroid model. (**A**) Images of the spheroids formation captured from time 0 days to 10 days through the Incucyte^®^ Spheroid Software Module. (**B**) The histograms showed the analysis of cell number, the area measurement and analysis of viability, respectively, in WT and ANXA1 KO spheroids. The analyses have been performed as reported in Material and Methods section. In particular, the number of cells has been calculated through the Countess Automated Cell Counter; the area by using Incucyte^®^ Spheroid Software Module and the luminescence has been measured with the CellTiterGlo. Data represent the mean of five independent experiments ± SD. * *p* < 0.05; *** *p* < 0.001.

**Figure 2 cancers-14-04764-f002:**
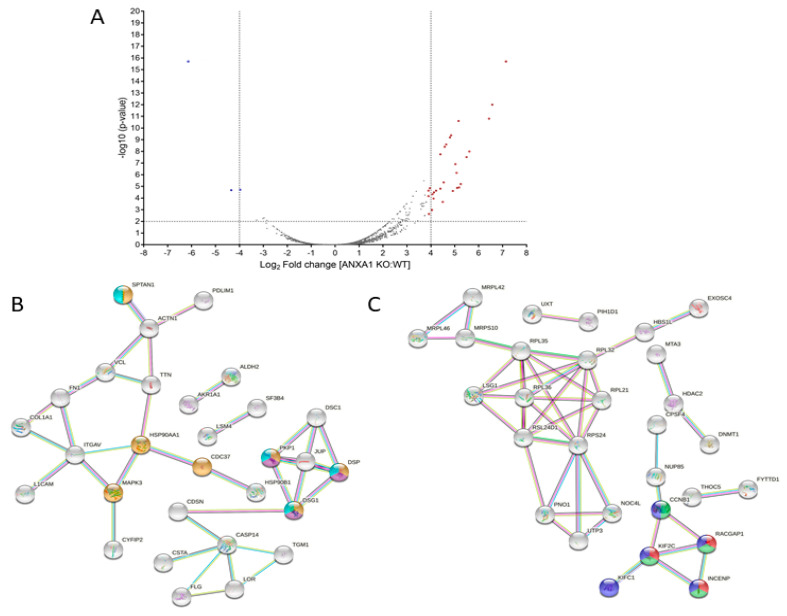
(**A**) Volcano plot of significance vs. fold change of ANXA1 KO MIA PaCa-2 vs. WT proteins. (**B**) STRING analysis of ANXA1 KO more abundant proteins annotated as being involved in programmed cell death (yellow) and apoptotic cleavage of cell adhesion proteins (violet and ciano) pathways. (**C**) STRING analysis of ANXA1 KO less abundant proteins annotated as being involved in mitotic sister chromatid segregation (green), spindle midzone assembly (red) and mitotic nuclear division pathways (blue).

**Figure 3 cancers-14-04764-f003:**
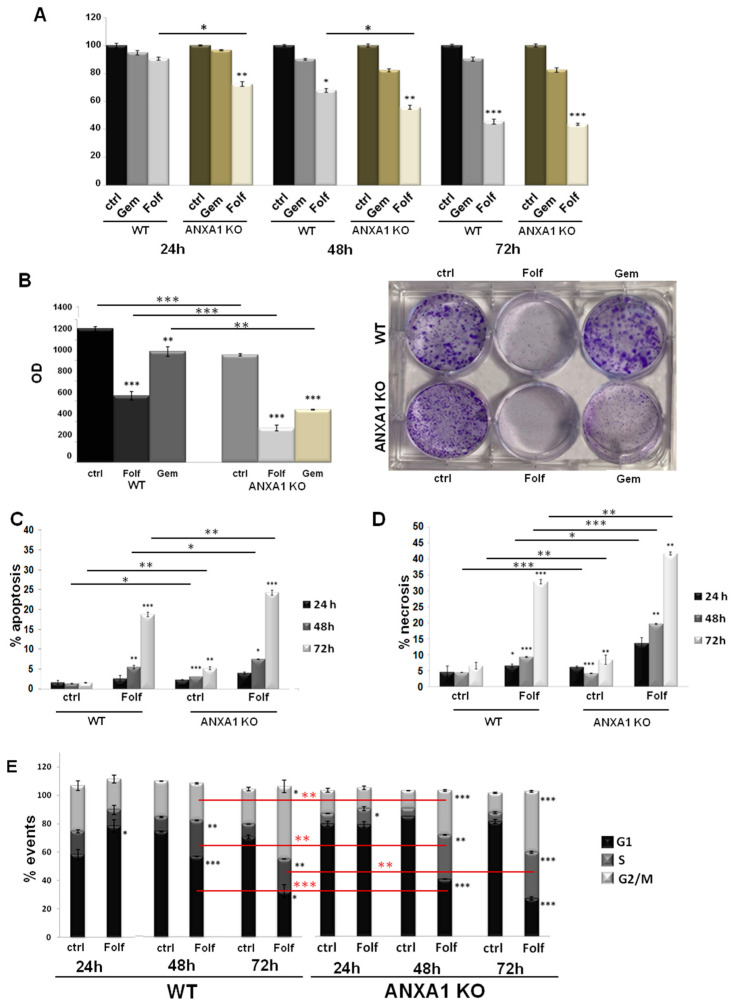
Analysis of drugs effects on WT and ANXA1 KO MIA PaCa-2 cells. (**A**) The histograms reported the mean of *n* = 5 MTT analysis at 24, 48 and 72 h on these cells treated or not with Gemcitabine 1 µM or FOLFIRINOX 25 µM. (**B**) Representative images of the clonogenic assay performed on WT and ANXA1 KO MIA PaCa-2 cells in presence or not of Gemcitabine 1 µM or FOLFIRINOX 25 µM for 24 h. The histograms refer to the optical density (OD) obtained from 1% SDS dissolution of cells and read to spectrophotometer. Apoptosis (**C**) and necrosis (**D**) analysis of WT and ANXA1 KO MIA PaCa-2 cells in presence or not of FOLFIRINOX (25 µM) after 24, 48 and 72 h. (**E**) WT and ANXA1 KO MIA PaCa-2 histograms reporting the number of events in different cell cycle phases after 24, 48 and 72 h of FOLFIRINOX (25 µM) treatment. These parameters have been assessed by staining cells with PI and through the Cell Quest program of FACScan cytometer. Cell cycle has been evaluated by a further analysis through ModFit LT software. Data represent the mean of five independent experiments ± SD. * *p* < 0.05; ** *p* < 0.01; *** *p* < 0.001 for Folf treated cells vs. control samples. ** *p* < 0.01; *** *p* < 0.001 on red lines for ANXA1 KO experimental points vs. WT.

**Figure 4 cancers-14-04764-f004:**
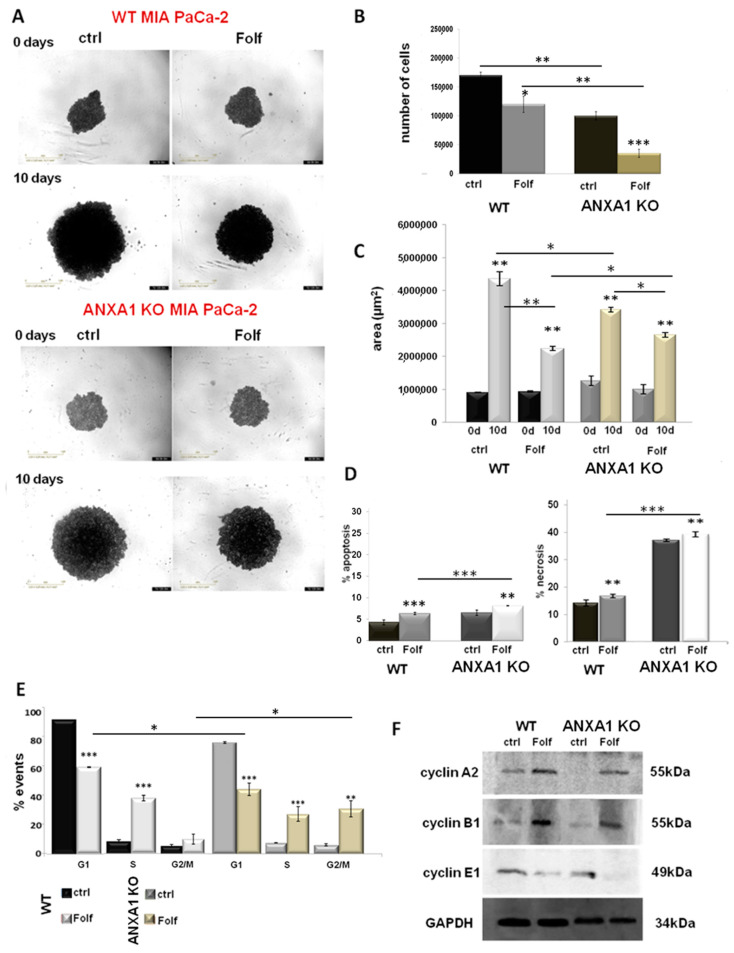
Analysis of Folf effects on WT and ANXA1 KO MIA PaCa-2 spheroids. (**A**) Spheroid growth images taken immediately after the seeding in plate and after the treatments performed as specified in Material and Methods section. Magnification 10×. Bar = 50 μm. The histograms showed the analysis of WT and ANXA1 KO spheroids by (**B**) count of cell number and (**C**) the area measurement, pre- and post treatment with Folf (d = day). These analyses have been performed by using the Countess Automated Cell Counter for the number of cells, the Incucyte^®^ Spheroid Software Module to obtained spheoird area and as the CellTiterGlo for the luminescence. (**D**) Apoptosis and necrosis and (**E**) cell cycle analysis of WT and ANXA1 KO spheroids after the Folf (25 µM) treatment obtained by readig PI stained samples at FACScan cytometer. (**F**) Western blot on protein extract from WT and ANXA1 KO MIA PaCa-2 spheroids analyzed for cyclin A2, cyclin B1, cyclin E1 and GAPDH, this latter used as housekeeping. The shown blots are representative of *n* = 3 experiments with similar results. Data represent the mean of five independent experiments ± SD. * *p* < 0.05; ** *p* < 0.01; *** *p* < 0.001. The uncropped blots are shown in Appendix A.

**Figure 5 cancers-14-04764-f005:**
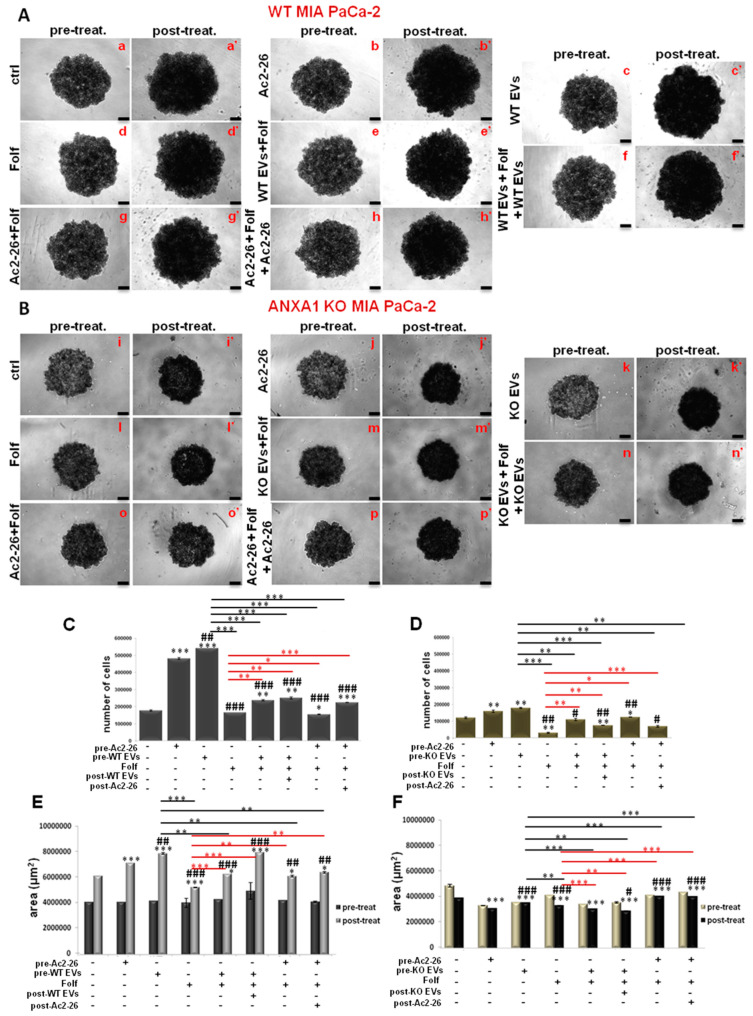
Analysis of autocrine EVs effects on drug treatment on WT and ANXA1 KO MIA PaCa-2 spheroids. WT (**A**) and ANXA1 KO (**B**) spheroids images taken at day 10 (panels a–h for WT and i–p for ANXA1 KO) and after the Folf (25 µM), EVs and Ac2-26 (1 µM) treatments, alone and in combination followed by further 3 days of maintenance in culture with fresh medium (panels a’–h’ for WT and i’–p’ for ANXA1 KO). Magnification 10×. Bar = 50 μm. The histograms show the analysis of WT (**C**–**E**) and ANXA1 KO (**D**–**F**) spheroids by count of cell number and the area measurement, pre- and post treatment, respectively. Data represent the mean of five independent experiments ± SD.* *p* < 0.05; ** *p* < 0.01; *** *p* < 0.001 for treated spheroids vs. control samples; # *p* < 0.05; ## *p* < 0.01; ### *p* < 0.001 for 0.001 for experimental points vs. Ac2-26 samples; * *p* < 0.05; ** *p* < 0.01; *** *p* < 0.001 on red lines for experimental points vs. Folf treatment; ** *p* < 0.01; *** *p* < 0.001 on black lines for experimental points vs. EVs samples.

**Figure 6 cancers-14-04764-f006:**
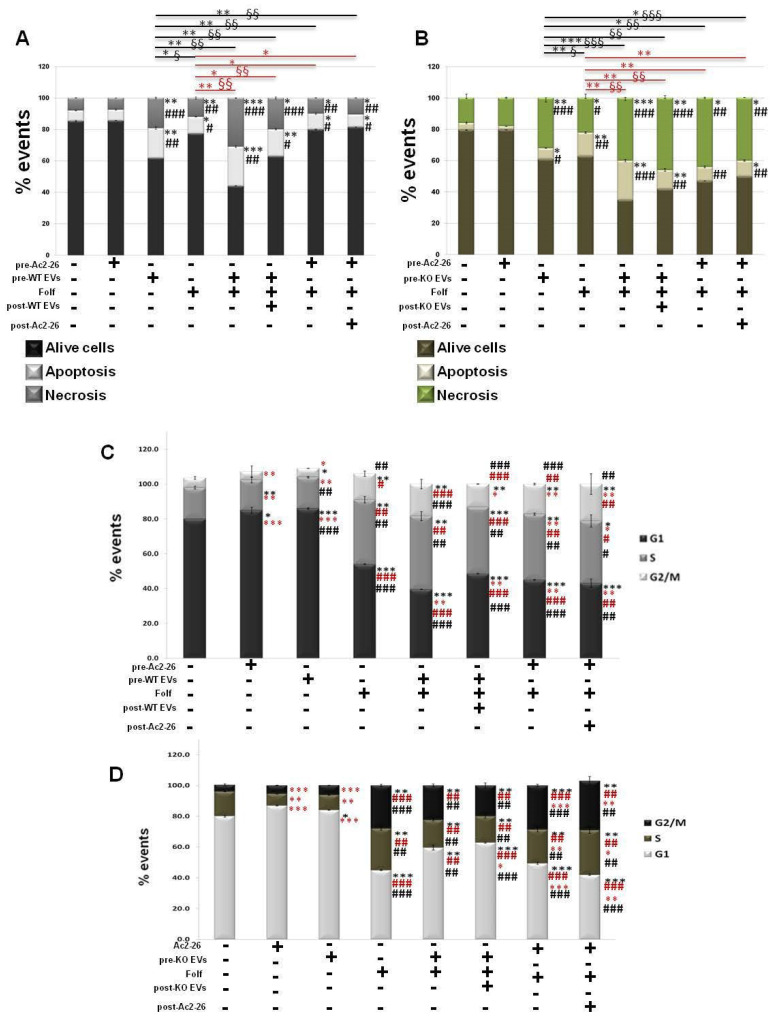
Analysis of autocrine EVs effects on drug treatment on cell death and cycle of WT and ANXA1 KO MIA PaCa-2 spheroids. The analysis was carried out after the Folf (25 µM), EVs and Ac2-26 (1 µM) treatments, alone and in combination followed by further 3 days of maintenance in culture with fresh medium. The histograms show the analysis of WT (**A**) and ANXA1 KO (**B**) in term of percentage of apotosis and necrosis evaluated by flow cytometry. Cell cycle assessment through ModFit LT software of (**C**) WT and (**D**) ANXA1 KO spheroids after the different kinds of treatments. Data represent the mean of five independent experiments ± SD. black * *p* < 0.05; ** *p* < 0.01; *** *p* < 0.001 for treated spheroids vs. control samples; black # *p* < 0.05; ## *p* < 0.01; ### *p* < 0.001 for experimental points vs. Ac2-26 samples; red * *p* < 0.05; ** *p* < 0.01; *** *p* < 0.001 for experimental points vs. Folf treatment (red * *p* < 0.05; ** *p* < 0.01 on red lines for experimental points vs. Folf treatment for apoptosis analysis and § *p* < 0.05; §§ *p* < 0.01; §§§ *p* < 0.001 for the related experimental necrosis points); red # *p* < 0.05; ## *p* < 0.01; ### *p* < 0.001 for experimental points vs. EVs samples (black * *p* < 0.05; ** *p* < 0.01; *** *p* < 0.001 on black lines for experimental points vs. Folf treatment for apoptosis analysis and § *p* < 0.05; §§ *p* < 0.01; §§§ *p* < 0.001 for the related experimental necrosis points).

## Data Availability

The data presented in this study are available upon request from the corresponding author.

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
