# Peer review of "Role of Intracellular and Extracellular Annexin A1 in MIA PaCa-2 Spheroids Formation and Drug Sensitivity"

_cancers, 2022, doi:10.3390/cancers14194764_

Round 1

Reviewer 1 Report

The authors used a human lineage of epithelial cells from pancreatic carcinoma (Mia PaCa-2) to develop spheroids. Additionally, AnxA1-depleted (KO) Mia PaCa-2 cells, obtained by CRISPPR/Cas9 genome editing system, have been established to create spheroids. Spheroids were employed to investigate the role of chemotherapy (5-fluoruracil, irinotecan, and oxaliplatin associated or not with gemcitabine and NAB-paclitaxel) and tumor-deriving extracellular vesicles on tumor development and growth. As it has been previously demonstrated, downregulating AnxA1 is connected to the chemoresistance in pancreatic cancer, the AnxA1-KO Mia PaCa-2-derived spheroids seem to be a novel and promising tool for pre-clinical investigations. Indeed, tumor spheroids have emerged to mimic in vivo structure and cellular interactions in basic research, drug screening, and preclinical studies. Specifically, different lineages of pancreatic tumor cells have been employed for this purpose. Although the instability and disaggregation of Mia-PaCa-based spheroids mainly during manipulation have been related, there are some data showing promising protocols improving these characteristics. 

Many experiments were performed, materials and methods were organized, and the manuscript presented many relevant results. Adequate references are used.

The novelty of the Novizio and cols. manuscript/data is AnxA1 KO Mia-PaCa-derived spheroids.  In this regard, although the proteomic profile was assessed, a discrete characterization was made (cell number, area, and cell viability). The authors emphasize that AnxA1 KO spheroids are an innovative model to understand molecular aspects of AnxA1 activity, as a different pattern of proteins was determined by proteomic analysis compared to WT spheroids and they are correct. The applicability of WT and AnxA KO spheroids were evaluated by treatment with different chemotherapy agents with the goal of testing drug sensitivity. In this context, tumor-derived extracellular vesicles (EV) and the AnxA1 peptide Ac2-26 were also employed. 

MAJOR REVISIONS:

EV and Ac2-26 were employed to clarify mechanisms by which AnxA1 modulates the tumor environment. However, some points should be addressed:

1.     Did the EVs content characterized? 

2.     Using EV in spheroids culture, and suggesting that EVs mainly content Anxa1, what is the hypothesis of AnxA1 actions? 

3.     There is no characterization regarding the receptors involved in AnxA1 and Ac2-26 actions. As AnxA1 acts by FPR2 interactions, was the expression of this receptor investigated? Is there any characterization regarding the expression and secretion of AnxA1 by Mia-PaCa cells used in the experimental approach? Was the expression profile of FPR1 and FPR2 assessed in Mia-PaCa? Did expression of those receptors change when cells were AnxA1 knockout? 

4.     The authors mentioned that area and compaction of AnxA1 spheroids were observed in comparison to WT spheroids. Is this a direct effect of the absence of AnxA1?

5.     What is the importance of measuring the spheroid area?  

6.     Investigation of spheroid cell viability was performed using CellTiterGlo. Does this protocol allow to an investigation of viability inside of spheroids?

7.     At 2. Material and Methods section: Most methodologies employed in the work were better described than flow cytometry (2.10 Flow cytometry cell cycle and cell death). Flow cytometry was used for cell cycle and cell death (apoptosis and necrosis). Also, it has cited that propidium iodide was employed. How was apoptosis investigated? How many events were considered for each analysis? It should be detailed.

8. In the results (lines 231 – 234) it was mentioned that macroscopically was observed a compact spheroid with an outer proliferative layer and a hypoxic and necrotic core. How is it possible to analyze proliferation, hypoxia, and necrosis macroscopically? 

9. In some figures, such as Fig3 – C, D presented a precarious scale at the Y axis, impairing correct data evaluation. It should be improved. 

10. In the result graphics there are too many symbols indicating statistical analyses. Excess symbols compromise the quality of figures. It is very difficult to understand. This is the case in Fig 3 – F, Fig 5 – C, D, E, F, and Fig 6. I suggest re-think a better way to organize it. The same is observed in spheroids images when cells are treated with Ac2-26+Folf+AC2-26 or WTEVs+Folf+WTEVs….

11.  Results section: Figure 4B – Pg 10 – Lines 313 – 315: The authors stated “On the other hand, AnxA1 KO spheroids, after Folf treatment, markedly showed an important reduction in both size and cell number (Fig. 4B and C)…” However, observing Figure 4, the cell numbers for WT and KO before and after Folf treatment seem to be equivalent. Also, the magnitude of modifications and cell behavior of AnxA1 KO is similar to WT. Could you explain?

12.  For Fig. 4F (pa 10, lines 327 – 328) The authors described that “…cyclin B1 showed a more evident signal in WT experimental point with Folf than in the related AnxA KO one”. Is this relevant? Moreover, in Figure 4F, the pattern of A2, B1, and E1 cyclin expressions seem equivalent in WT and AnXA1 spheroids. The difference is only observed in the supplementary data where Cyclin A2 and B1 expressions are improved in AnxA1-derived spheroids after Folf treatments. The relevance of data should be highlighted. Take care of excessive descriptions of all changes observed, as they may mask relevant results and difficult identify important/critical findings.

13.  A positive control for necrosis and apoptosis is missed.

Minor revisions:

1.     Introduction is well written, and the spheroid protocol is emphasized. Although the authors mentioned the role of AnxA1 and EV on cancer, they missed the role of Ac2-26.  

2.     Standardize nomenclature. For instance, sometimes is used AnxA1 and sometimes ANAX1; Folf, and FOLFIRINOX.

3.     At 2. Material and Methods: Western blotting - secondary antibodies were missed.

Author Response

#1

The authors used a human lineage of epithelial cells from pancreatic carcinoma (Mia PaCa-2) to develop spheroids. Additionally, AnxA1-depleted (KO) Mia PaCa-2 cells, obtained by CRISPPR/Cas9 genome editing system, have been established to create spheroids. Spheroids were employed to investigate the role of chemotherapy (5-fluoruracil, irinotecan, and oxaliplatin associated or not with gemcitabine and NAB-paclitaxel) and tumor-deriving extracellular vesicles on tumor development and growth. As it has been previously demonstrated, downregulating AnxA1 is connected to the chemoresistance in pancreatic cancer, the AnxA1-KO Mia PaCa-2-derived spheroids seem to be a novel and promising tool for pre-clinical investigations. Indeed, tumor spheroids have emerged to mimic in vivo structure and cellular interactions in basic research, drug screening, and preclinical studies. Specifically, different lineages of pancreatic tumor cells have been employed for this purpose. Although the instability and disaggregation of Mia-PaCa-based spheroids mainly during manipulation have been related, there are some data showing promising protocols improving these characteristics. 

Many experiments were performed, materials and methods were organized, and the manuscript presented many relevant results. Adequate references are used.

The novelty of the Novizio and cols. manuscript/data is AnxA1 KO Mia-PaCa-derived spheroids.  In this regard, although the proteomic profile was assessed, a discrete characterization was made (cell number, area, and cell viability). The authors emphasize that AnxA1 KO spheroids are an innovative model to understand molecular aspects of AnxA1 activity, as a different pattern of proteins was determined by proteomic analysis compared to WT spheroids and they are correct. The applicability of WT and AnxA KO spheroids were evaluated by treatment with different chemotherapy agents with the goal of testing drug sensitivity. In this context, tumor-derived extracellular vesicles (EV) and the AnxA1 peptide Ac2-26 were also employed. 

MAJOR REVISIONS:

EV and Ac2-26 were employed to clarify mechanisms by which AnxA1 modulates the tumor environment. However, some points should be addressed:

  1. Did the EVs content characterized? 

Authors’ answer

Following the reviewer’s suggestion, we specified how we have characterized the EVs content which we routinely obtain from both WT and ANXA1 KO MIA PaCa-2 cells. In particular, we clarified that the purified EVs have been examined first by Field Emission Scanning Electron Microscopy (FE-SEM) and by dynamic light-scattering (DLS). Two techniques by which typical rounded particles ranging from 30 to 180 nm in diameter have been observed. In addition, through Western blot analysis we have confirmed the presence of the proteins TSG101, CD63 and CD81 as they are the main characterized and the mostly used EVs markers, together with the absence of the calreticulin, generally utilized as negative control since it is exposed in the surface of apoptotic cells. These information, previously demonstrated as reported in references #16 and #17, have been included in the paragraph 3.5 of the Results section.

  1. Using EV in spheroids culture, and suggesting that EVs mainly content Anxa1, what is the hypothesis of AnxA1 actions? 

Authors’ answer

We thank the reviewer for point this out. On the spheroids, as well as on the monolayer model we have previously demonstrated, ANXA1 as protein element of the EVs membranes can act binding FPR1 and FPR2 which are described to be the most characterized ANXA1 receptor partners. In particular, we have characterized in depth the expression of these receptors on MIA PaCa-2 cells surface (see references #14, 17 and 40) both in presence and in absence of ANXA1. Actually, the effects we have found in this study as mediated by WT EVs and Ac2-26, more than ANXA1 KO EVs, can confirm this issue since it has been assessed that the cascade of reaction triggered by FPR activation, is able to induce PC promotion. Please also refer to the response to comment #3.

  1. There is no characterization regarding the receptors involved in AnxA1 and Ac2-26 actions. As AnxA1 acts by FPR2 interactions, was the expression of this receptor investigated? Is there any characterization regarding the expression and secretion of AnxA1 by Mia-PaCa cells used in the experimental approach? Was the expression profile of FPR1 and FPR2 assessed in Mia-PaCa? Did expression of those receptors change when cells were AnxA1 knockout? 

Authors’ answer

The expression of FPR1 and FPR2 has been previously characterized in our published works and it does not change in ANXA1 KO MIA PaCa-2 compared to the WT counterpart as we stated through flow cytometry and RT-PCR in references #14, 17 and 40. Thus, as described in this previous works, we could suggest that extracellular ANXA1 binds these receptors triggering their activation and the following intracellular cascade of reactions. We have now clarified this aspect in the Discussion section of the revised version of the manuscript (page 21 of 25).

  1. The authors mentioned that area and compaction of AnxA1 spheroids were observed in comparison to WT spheroids. Is this a direct effect of the absence of AnxA1?

Authors’ answer

We thank the reviewer who gave us the opportunity to better explain the features of ANXA1 KO spheroids compared to WT ones. In detail, we have demonstrated that ANXA1 KO MIA PaCa-2 cells have been not particularly adept at forming the spheroids as 3D model. Indeed these latter, according to published literature protocols, have displayed an unstable structure comparable to an "aggregate" with less tightly packed cells. These aggregates do not show a strongly compactness as for the compact spherical cultures like spheroids showing in WT ones. Moreover, our analysis about the structure, the area and the number of cells have highlighted the ability of WT cells to form spheroids compared to ANXA1 KO ones, allowing us in turn to suggest that one of the key elements for this differences can just be the protein of our interest. Furthermore, as reported in [14], the presence of intracellular ANXA1 is able to preserve the cytoskeleton integrity and to maintain a malignant phenotype, capabilities which could also trigger the greater stability and compactness of WT spheroid. This aspect has been now detailed in the Discussion section of the revised manuscript (page 19 of 25).

  1. What is the importance of measuring the spheroid area?  

Authors’ answer

In order to obtain the reproducible and compact spheroids we must take into account several specific aspects of this 3D cell model. The physiological state of spheroids strongly depends on the spheroid size, the specific kind of tumor cells composing the model, the cell density, the number of cells and finally the culture time. Indeed the measurement of the spheroid area generates information closely correlated to this physiological state and its growth in order to form the structure miming the tumor [see the new added reference 53]. To date, it is possible to measure the area with the specific software “Incucyte® Spheroid Software Module” correlated and included in “Incucyte® Live-Cell Analysis System” that provided to monitor the spheroid formation by taking photos every 6 hours repeat scanning.

  1. Investigation of spheroid cell viability was performed using CellTiterGlo. Does this protocol allow to an investigation of viability inside of spheroids?

Authors’ answer

In our work, we have used CellTiterGlo to analyze the spheroid cell viability. As reported in the instructions and specifications, the “CellTiter-Glo® Luminescent Cell Viability” kit is a homogeneous method of determining the number of viable cells in culture based on quantization of the ATP present, an indicator of metabolically active cells [https://ita.promega.com/products/cell-health-assays/cell-viability-and-cytotoxicity-assays/celltiter_glo-luminescent-cell-viability-assay/?catNum=G7570]. Moreover, this kid of measurement is also suggested in the application note of IncuCyte Live-Cell Analysis system that explains to use the ATP viability assay of CellTiter-Glo in order to study the generate ATP relative luminescent units (RLU) values. As reported in previous works, now included in the manuscript as #41; 42; 43 references, this feature means the change in the cell metabolism upon 3D culture formation, particularly once the spheroid becomes large and develops a hypoxic core.

  1. At 2. Material and Methods section: Most methodologies employed in the work were better described than flow cytometry (2.10 Flow cytometry cell cycle and cell death). Flow cytometry was used for cell cycle and cell death (apoptosis and necrosis). Also, it has cited that propidium iodide was employed. How was apoptosis investigated? How many events were considered for each analysis? It should be detailed.

Authors’ answer

According to the reviewer’s comment, we improved the paragraph 2.10. of the Material and methods section. In particular, we detailed the procedures by which we have evaluated the apoptosis/necrosis and cell cycle. Please refer to page 5 of 25.

  1. In the results (lines 231 – 234) it was mentioned that macroscopically was observed a compact spheroid with an outer proliferative layer and a hypoxic and necrotic core. How is it possible to analyze proliferation, hypoxia, and necrosis macroscopically? 

Authors’ answer

The measurement of spheroid morphology has been found to be a key parameter in experimental standardization. Several studies have reported its characteristic layer-like structure: a necrotic core, an inner layer of quiescent cells, and a layer of proliferating cells that is just like the real tumor structure [reference #52 in the revised manuscript; Ishiguro T, Ohata H, Sato A, Yamawaki K, Enomoto T, Okamoto K. Tumor-derived spheroids: Relevance to cancer stem cells and clinical applications. Cancer Sci. 2017; 108(3):283-289. doi: 10.1111/cas.13155; Imamura Y, Mukohara T, Shimono Y, Funakoshi Y, Chayahara N, Toyoda M, Kiyota N, Takao S, Kono S, Nakatsura T, Minami H. Comparison of 2D- and 3D-culture models as drug-testing platforms in breast cancer. Oncol Rep. 2015; 33(4):1837-43. doi: 10.3892/or.2015.3767; and Ref #31 already included in the manuscript]. This necrotic core is formed due to the limitation of nutrient transportation to the middle region respect the outliers areas where cells remain more able to proliferate. These parameters (i.e., diameter, volume, sphericity, etc.) can be obtained through various types of microscopy, such as light, fluorescence, and light sheet microscopes [Sant S, Johnston PA. The production of 3D tumor spheroids for cancer drug discovery. Drug Discov Today Technol. 2017; 23:27-36. doi: 10.1016/j.ddtec.2017.03.002.]. Thus, the acquisition of images to these instruments still represents the most effective control and monitoring of spheroid morphology to understand the spheroid growth and features, particularly seeing a darker central area and jagged edges. Moreover, thanks to advances in scientific research, to date there are some tools which are able to monitor growth and analyze the size and volume of the spheroid and its morphology. In detail, in our study, as reported in Material and method section, to monitor the spheroid formation and growth we used the Incucyte® Live-Cell Analysis System by acquiring 6 hours repeat scanning images, up to the end. Finally, the spheroid analysis of each parameter has been performed thanks to Incucyte® Spheroid Software Module through DF-Brightfield images obtained in the real time. We have now clarified this concern in the revised version of the manuscript (please see page 6 of 25)  and a further reference has been included (#52).

  1. In some figures, such as Fig3 – C, D presented a precarious scale at the Y axis, impairing correct data evaluation. It should be improved. 

Authors’ answer

In agreement with the reviewer’s suggestion, we modified the Y axis of the indicated histogram in order to make it clear.

  1. In the result graphics there are too many symbols indicating statistical analyses. Excess symbols compromise the quality of figures. It is very difficult to understand. This is the case in Fig 3 – F, Fig 5 – C, D, E, F, and Fig 6. I suggest re-think a better way to organize it. The same is observed in spheroids images when cells are treated with Ac2-26+Folf+AC2-26 or WTEVs+Folf+WTEVs….

Authors’ answer

As suggested by the reviewer, we reorganized the significance symbols in each histogram in figure 3, 4, 5, 6. Accordingly, also the related figure legends have been modified. Moreover, we adjusted the label of spheroids images in figure 5 in order to make easier the reading.

  1. Results section: Figure 4B – Pg 10 – Lines 313 – 315: The authors stated “On the other hand, AnxA1 KO spheroids, after Folf treatment, markedly showed an important reduction in both size and cell number (Fig. 4B and C)…” However, observing Figure 4, the cell numbers for WT and KO before and after Folf treatment seem to be equivalent. Also, the magnitude of modifications and cell behavior of AnxA1 KO is similar to WT. Could you explain?

Authors’ answer

We apologize for the unexpected mistake regarding the upload of figure 4B. Indeed, we uploaded in the first version of the manuscript a wrong histogram which did not reflect our data. The new version of the figure, we have now modified, reports the right assessment we obtained about the number of cells calculated for WT vs ANXA1 KO spheroids in absence (as it is confirmed in figures 1B and 5 C/D) and in presence of FOLFIRINOX. In detail, drug treatment induced a significant decreased in the number of cells showing a greater effect of the ANXA1 KO spheroids compared the WT ones. About figure 4C, following FOLFIRINOX treatment, the ANXA1 KO spheroids area has been markedly affected by the appearance of a notable amount of empty spaces due to the lost of compactness, likely derived from the loss of cell-cell interactions. This observation has been better defined in the paragraph 3.4 of the Results section in order to better reflect the data shown in the related figure. Thus, we really thank the reviewer to have allowed us to clarify this important issue.

  1. For Fig. 4F (pa 10, lines 327 – 328) The authors described that “…cyclin B1 showed a more evident signal in WT experimental point with Folf than in the related AnxA KO one”. Is this relevant? Moreover, in Figure 4F, the pattern of A2, B1, and E1 cyclin expressions seem equivalent in WT and AnXA1 spheroids. The difference is only observed in the supplementary data where Cyclin A2 and B1 expressions are improved in AnxA1-derived spheroids after Folf treatments. The relevance of data should be highlighted. Take care of excessive descriptions of all changes observed, as they may mask relevant results and difficult identify important/critical findings.

Authors’ answer

In accordance with the reviewer’s comment, we improved the description of the results shown in Figure 4F. In particular, we have highlighted that the analysis of the single cyclin expression, resulted relevant just as confirmation of the data concerning the alteration of the cell cycle in the presence of the drug. Moreover, we clarified the different levels of cyclins expression not only between treated WT vs ANXA1 KO spheroids but also the treated vs not treated experimental points for each cell line. Indeed, since the analyzed samples could be difficult to directly compare all together, we confirmed the Western blot images through the densitometry analysis which highlighted the differences among protein levels. Moreover, we underlined in the 3.4 paragraph of the Results section the importance of these finding which strongly make relevant the different kind of variation of cell cycle phases in the analyzed spheroids.

  1. A positive control for necrosis and apoptosis is missed.

Authors’ answer

We did not included a positive control for the analysis of apoptosis and necrosis. But we used gemcitabine in all these experiments. Treatment with gemcitabine confirms the absence of a significant level of cell death in accordance with our previous studies for which MIA PaCa-2 cells have shown a significant resistance to gemcitabine 1µM administered for 24 hours [Boccia E, Alfieri M, Belvedere R, Santoro V, Colella M, Del Gaudio P, Moros M, Dal Piaz F, Petrella A, Leone A, Ambrosone A. Plant hairy roots for the production of extracellular vesicles with antitumor bioactivity. Commun Biol. 2022; 5(1):848. doi: 10.1038/s42003-022-03781-3]. We have not showed these data in this work, but we have now included this in the Material and Methods section of the revised manuscript (page 5 of 25).

Minor revisions:

  1. Introduction is well written, and the spheroid protocol is emphasized. Although the authors mentioned the role of AnxA1 and EV on cancer, they missed the role of Ac2-26.  

Authors’ answer

According to the reviewer’s comment, we included information about Ac2-26 in the Introduction section. In this regard, two new references (#38 and #39) have been included (Page 3 of 25).

  1. Standardize nomenclature. For instance, sometimes is used AnxA1 and sometimes ANAX1; Folf, and FOLFIRINOX.

Authors’ answer

We apologize for the unclear description. We have now corrected the term ANAX1 in figure 4 with the right form ANXA1. About Folf, we only used the acronym of the FOLFIRINOX in figures since it results easier to write. We highlighted it in paragraph 2.7. of Material and methods section.

  1. At 2. Material and Methods: Western blotting - secondary antibodies were missed.

Authors’ answer

The secondary antibodies have been now included.

Reviewer 2 Report

General comments:

The authors establish spheroid cultures using pancreatic cell lines and subsequently compare how Annexin A1 KO affects spheroid growth, drug sensitivity in 2D and 3D models, and drug sensitivity in precense of extracellular vehicles. The conclusions of this study is somewhat unclear and the paper lacks any mechanistic insights into the role of Annexin A1 in spheroid formation and dug sensitivity which in my view limits the impact of this study.

The authors should consider major revisions and rewriting of this manuscript before any resubmission. Throughout the manuscript data is described very selectively resulting in questionable and misleading conclusions. The manuscript doesn't flow, and figures are often difficult to read and insufficiently described in legends. In their figures the authors should consider reducing the number of colors and instead use shading or omit colors where sample labelling alone is sufficient. There’s no need for coloring bars in plots describing WT cells differently from bars in ANXA1 KO if data presented in different plots/figure panels.

Specific comments:

1. The angle of this paper seems to advocate the use of spheroid compared to 2D culture models as others have previously shown to better reflect certain aspects of tumor biology. Yet, it seems to me that the data presented in this manuscript show similar conclusions from 2D vs 3D cultures (Fig. 4&5). The authors should discuss this and specifically compare the results of Figs. 3 & 4 rather than describing those independently. Maybe the two figures should be merged to allow side-by-side comparison.

2. Related to the above, while the authors show that different time points can lead to different conclusions in their 2D model, they only show a single time point (which is not specified) for the spheroid model. Please provide a time course for spheroids as well or clearly specify why the chosen single time point is justified.

3. The authors could consider the orders of figures presented to avoid having to switch back and forth between 2D vs 3D models

4. It is unclear what cut off the authors have used for their mass spec analysis. In the text a ratio of 2/0.5 are mentioned but in Fig. 4 LFC of 4 are marked. Please clarify 

5. The STRING analysis data are very selectively described in the manuscript. All three colored pathways can be found in up- and down-regulated proteins, yet the conclusions in the text draw a very different picture. Also, the picture quality of the STRING data is rather bad and proteins are hardly readable. Please provide better illustration.

6. In addition to the STRING protein-protein analysis the authors should perform a GO-term enrichment analysis for their proteomics analysis independently of protein interaction data.

7. Figure 3 B: bar plots and plate picture does not agree with each other. Is there a labelling mistake here? If that's the case the authors should very carefully check the rest of their figures and data for similar problems.

8. Throughout the manuscript the authors should clearly state whether experiments in WT and ANXA1 KO result in different effects or same trends. The main text of implies differences while figures show that drug treatment have same effect in WT and ANXA1 KOs.

9. The authors have to describe what Ac2-26 is and how it is being used in their experiments. This appears out of the blue without any introduction.

10. The authors should also better introduce Folfirinox and the authors drugs used in this study. What’s their target and mode of action? How is this connected to Annexin A1? Also is a rather high concentration of 25 µM standard in cell culture experiments?

11. For cell cycle analysis it may be more intuitive for readers if they label G2 as G2/M as the two phases are hard to tell apart from each other.

12. Figure legends (e.g. Figure 3) should be more detailed and should describe which assay was used, treatment details, color keys and stats.

13. Instead of using different symbols for statistical comparison between different groups (*, #,@,c, etc.) it would less confusing for readers if the authors stick to one symbol (*) and draw lines above bars connecting samples being compared. Also, did the authors correct stats for multiple testing?

14. Several methods sections are missing including: apoptosis & necrosis assays, STRING analysis

15. Line 240: the authors present no data supporting their statement of cellular metabolic activity. Please adjust text accordingly and stick as close as possible to facts supported in figures rather than making unsupported bold statements (several other examples exist throughout the manuscript).

Author Response

#2

General comments:

The authors establish spheroid cultures using pancreatic cell lines and subsequently compare how Annexin A1 KO affects spheroid growth, drug sensitivity in 2D and 3D models, and drug sensitivity in presence of extracellular vehicles. The conclusions of this study is somewhat unclear and the paper lacks any mechanistic insights into the role of Annexin A1 in spheroid formation and dug sensitivity which in my view limits the impact of this study.

The authors should consider major revisions and rewriting of this manuscript before any resubmission. Throughout the manuscript data is described very selectively resulting in questionable and misleading conclusions. The manuscript doesn't flow, and figures are often difficult to read and insufficiently described in legends. In their figures the authors should consider reducing the number of colors and instead use shading or omit colors where sample labelling alone is sufficient. There’s no need for coloring bars in plots describing WT cells differently from bars in ANXA1 KO if data presented in different plots/figure panels.

Authors’ answer

Taking advantage from these suggestions, we strongly modified overall the manuscript. In particular, we enriched the figures legends of more details; we also readapted all figures eliminating the colors of the plots and making them easier to be interpreted. Moreover the changes we have performed have been specified in the following answers and highlighted in the tracked change version of the manuscript. Regarding the role of Annexin A1 in spheroid formation and dug sensitivity we have now included further details. We appreciate the insightful comments of the reviewer and have modified the manuscript accordingly.

Specific comments:

  1. The angle of this paper seems to advocate the use of spheroid compared to 2D culture models as others have previously shown to better reflect certain aspects of tumor biology. Yet, it seems to me that the data presented in this manuscript show similar conclusions from 2D vs 3D cultures (Fig. 4&5). The authors should discuss this and specifically compare the results of Figs. 3 & 4 rather than describing those independently. Maybe the two figures should be merged to allow side-by-side comparison.

Authors’ answer

We have chosen the 3D cultures because the spheroids generated with cancer cell lines mimic tumor architecture and share its limited drug penetration properties since drugs are largely confined to the outer cell layers [Desoize B, Jardillier J. Multicellular resistance: a paradigm for clinical resistance? Crit Rev Oncol Hematol. 2000; 36(2-3):193-207. doi: 10.1016/s1040-8428(00)00086-x]. Moreover, the spheroids with increasing size, inward proliferation due to oxygen/nutrient gradients, as well as a potential accumulation of catabolites in central regions, can highly reminiscent of poorly vascularized areas in solid tumors. All these features affect the protein expression, the distribution and function of biomodulators and also the penetration, binding and bioactivity of therapeutic drugs. According to literature data, we might have expected that many drug candidates could lose efficacy in the 3D models [Friedrich J, Eder W, Castaneda J, Doss M, Huber E, Ebner R, Kunz-Schughart LA. A reliable tool to determine cell viability in complex 3-d culture: the acid phosphatase assay. J Biomol Screen. 2007; 12(7):925-37. doi: 10.1177/1087057107306839. Erratum in: J Biomol Screen. 2007; 12(8):1115-9; Mueller-Klieser W. Multicellular spheroids. A review on cellular aggregates in cancer research. J Cancer Res Clin Oncol. 1987;113(2):101-22. doi: 10.1007/BF00391431; Mueller-Klieser W. Three-dimensional cell cultures: from molecular mechanisms to clinical applications. Am J Physiol. 1997; 273(4):C1109-23. doi: 10.1152/ajpcell.1997.273.4.C1109; Mueller-Klieser W. Tumor biology and experimental therapeutics. Crit Rev Oncol Hematol. 2000; 36(2-3):123-39. doi: 10.1016/s1040-8428(00)00082-2]. On the other hand, there is also experimental evidence demonstrating that several drugs may exclusively be effective in 3D but not 2D cultures, as it has been seen in target-specific treatment modalities for which the molecular target is expressed only or particularly in a 3D environment [ #65; 66; 67 newly added references]. Accordingly to these concerns, in our case it has been necessary to test drug efficacy firstly by classical monolayer test assays, showing the different sensitivity of WT and ANXA1 KO MIA PaCa-2, and then on more complex spheroid structures. Thus, the finding of the similar trend in the response to FOLFIRINOX allowed us to confirm our results by taking advantage of 2D and 3D tumor models.

Furthermore, despite the huge effort on research and development of chemotherapies for pancreatic cancer, there are only a few treatment options approved in clinic to prolong the survival of patients. In particular, based on the emergence of the acquired resistance to gemcitabine, first used as first-line chemotherapy drug of choice, it has been necessary to resort to other drugs, such as FOLFIRINOX, both alone and in combination with gemcitabine in order to improve the patients outcome. In this scenario, the research challenges have also provided the use of the spheroids as a tool to critically contribute to a reduction in animal tests and to optimize drug candidates selection for enhanced tissue distribution and efficacy [#53 as new reference] This specific model can be considered an interesting upgrade, compared to the monolayer in vitro cultures, to better study the phenomenon of drug resistance since it preserves features with intermediated complexity, tending more towards the in vivo conditions [Kunz-Schughart LA, Freyer JP, Hofstaedter F, Ebner R. The use of 3-D cultures for high-throughput screening: the multicellular spheroid model. J Biomol Screen. 2004; 9(4):273-85. doi: 10.1177/1087057104265040; Kunz-Schughart LA. Multicellular tumor spheroids: intermediates between monolayer culture and in vivo tumor. Cell Biol Int. 1999; 23(3):157-61. doi: 10.1006/cbir.1999.0384; Yang TM, Barbone D, Fennell DA, Broaddus VC. Bcl-2 family proteins contribute to apoptotic resistance in lung cancer multicellular spheroids. Am J Respir Cell Mol Biol. 2009; 41:14–23. doi: 10.1165/rcmb.2008-0320OC; Hirschhaeuser F, Menne H, Dittfeld C, West J, Mueller-Klieser W, Kunz-Schughart LA. Multicellular tumor spheroids: an underestimated tool is catching up again. J Biotechnol. 2010; 148:3–15. doi: 10.1016/j.jbiotec.2010.01.012; Yamada KM, Cukierman E. Modeling tissue morphogenesis and cancer in 3D. Cell. 2007; 130:601–10. doi: 10.1016/j.cell.2007.08.006]. For all those reason, we chose to study the FOLFIRINOX effects first on WT and ANXA1 KO cell lines seeded in monolayer, then organized in spheroids despite the ability of these cells to arrange them in a well structured and reproducible 3D architecture. In addition, we had no information about the impact of ANXA1 in the response to drug treatment and if and/or how this activity could differentiate between the two kinds of models. These arguments have been reported in Discussion section in order to clarify our need to use both 2D and 3D systems (please refer to page 20 of 25).

  1. Related to the above, while the authors show that different time points can lead to different conclusions in their 2D model, they only show a single time point (which is not specified) for the spheroid model. Please provide a time course for spheroids as well or clearly specify why the chosen single time point is justified.

Authors’ answer

We thank the reviewer for having raised this particular aspect, concerning the choice of experimental time points. For 2D cultures, as usual, we tested the drug at 3 experimental times, 24, 48 and 72 hours, showing greater sensitivity at 24 and 48 hours by MTT assay. As for the spheroids, the choice of the specific experimental time was of fundamental importance because it generally depends by the physiological state of spheroids. This latter clearly is based on the spheroid size, the individual and cell type–specific behavior of the tumor cells, the cell density within the spheroid and also directly or indirectly on the culture time. Owing to literature data and our own experience, we have chosen 24 hours as the experimental time point in which we have the best conditions for which spheroids can be subsequently analyzed, preserving its structure and integrity. Moreover, it has been reported that in this 3D model, different kinds of molecules should be carefully checked in time <48 hours for testing [53]. Furthermore, the specific choice of 24 hours has also derived from the evidences obtained at the same experimental time point on the cellular monolayer by MTT assay. We relied more on this result than on the 2D necrotic one, since this feature represents an important variable in spheroids which are generally provided of a necrotic core, dependent on the cell line, packaging and cell growth. In second instance, we applied only one FOLFIRINOX treatment on spheroids, to later analyze them after 3 days, to unravel the delayed effect of FOLFIRINOX treatment on cancer cells since highest cell mortality occurs several days after treatment arrest [54]. We have now included these details in the 3.4 paragraph of the Results section of the revised manuscript.

  1. The authors could consider the orders of figures presented to avoid having to switch back and forth between 2D vs 3D models

Authors’ answer

As stated above, the importance to underline the potential differences in the cell sensitivity/resistance if they are disposed in monolayer or as spheroids led us to first assess this aspect on 2D model and only later on 3D counterpart. Taking into account that our work is focused on investigating the role of ANXA1 in MIA PaCa-2 spheroids, we have considered important to leave these two sets of experiments separately in order to highlight the similar trend of response to FOLFIRINOX, in presence or not of ANXA1.  

  1. It is unclear what cut off the authors have used for their mass spec analysis. In the text a ratio of 2/0.5 are mentioned but in Fig. 4 LFC of 4 are marked. Please clarify 

Authors’ answer

In Table S1, all differently expressed protein are reported considering up-regulated those proteins with a Fc ANXA1 KO vs WT ratio˃2 and down-regulated those proteins with a Fc ANXA1 KO vs WT ratio<0.5. These proteins are also depicted in Figure 2A where we marked Log2 of Fc ANXA1 KO vs WT ratio ˃4 and <-4, just for a pictorially purpose. Now it has been clarified both in the text and in the Table S1 in which we added a column reporting the Log2 of Fc ANXA1 KO vs WT.

  1. The STRING analysis data are very selectively described in the manuscript. All three colored pathways can be found in up- and down-regulated proteins, yet the conclusions in the text draw a very different picture. Also, the picture quality of the STRING data is rather bad and proteins are hardly readable. Please provide better illustration.

Authors’ answer

We provided better illustration; we also have changed colors, diversifying those for the up and down regulated proteins as now reported both in the text and in the Figure 2B and C caption.

  1. In addition to the STRING protein-protein analysis the authors should perform a GO-term enrichment analysis for their proteomics analysis independently of protein interaction data.

Authors’ answer

In Table S1 it is now reported a GO-term enrichment analysis for both GO and biological processes. In red we have highlighted all the terms related to apoptosis and cell death, whereas in green all the terms related to mitosis and its spindle.

  1. Figure 3 B: bar plots and plate picture does not agree with each other. Is there a labelling mistake here? If that's the case the authors should very carefully check the rest of their figures and data for similar problems.

Authors’ answer

We apologize for the mistake. We have now corrected the figure 3 indicated by the reviewer. We overall revised all figures; in particular, this one has been modified by eliminating the F panel in order to make easier the interpretation of results.

  1. Throughout the manuscript the authors should clearly state whether experiments in WT and ANXA1 KO result in different effects or same trends. The main text of implies differences while figures show that drug treatment have same effect in WT and ANXA1 KOs.

Authors’ answer

We thank the reviewer for this suggestion. Accordingly, we have modified the statements for results regarding the response to FOLFIRINOX drug treatment. Therefore, in figure 4 we first assessed the number of cells in WT and ANXA1 KO spheroids finding that it notably decreased in presence of FOLFIRINOX and this difference appeared more evident in absence of ANXA1. Accordingly, we here specify that, taking advantage of the comment #11 of the reviewer #1, we now replaced the histogram in figure 4A, wrongly uploaded in the first version of the manuscript, with a new one which rightly shows the analysis of cell number. Please also refer to the same results shown figures 1B and 5 C/D, already reported in the first version of our work. Then, about the evaluation of spheroids area, we improved the 3.4 paragraph of Results section highlighting that, in presence of FOLFIRINOX, this specific feature for ANXA1 KO spheroids, more than WT ones, appeared affected by a notable amount of empty spaces due to the lost of compactness, likely derived from the loss of cell-cell interactions. This observation confirms that not necessarily the number of cells and the measurement of size must be directly correlated. Additionally, the enhanced apoptosis and necrosis in response to FOLFIRINOX highlighted a same trend for WT and ANXA1 KO spheroids despite the higher percentage of death as a proper characteristic of models lacking ANXA1 with respect to the WT related controls. We have now clarified this aspect in the 3.4 paragraph. Next, we interestingly found a greater sensitivity of ANXA1 KO spheroids treated with FOLFIRINOX in the variation of cell cycle. We reported this outcome not only in figure 4E but also in 4F. This latter showed the different levels of cyclins. In particular, the increase of cyclins A2 and B1 after drug treatment appeared more significant for the ANXA1 KO samples (treated vs not treated) since this experimental point expressed a very low levels of both these proteins highlighting the poorer ability of these cells to proliferate in the 3D organization. On the other hand, the cyclin E1 increase appeared more evident for spheroids lacking ANXA1 due to the higher levels of this cyclin in not treated sample with respect to the WT counterpart. This aspect has been clarified in the revised version of the manuscript (please also refer to the response to the comment #12 of the reviewer #1). The response to the other treatments, performed with Ac2-26 and EVs, appeared significantly different between WT and ANXA1 KO spheroids confirming, as shown in our previous works (references #14-17, #19 and #40), the different level of aggressiveness that the two models maintain due to the presence or not of the protein of our interest.

  1. The authors have to describe what Ac2-26 is and how it is being used in their experiments. This appears out of the blue without any introduction.

Authors’ answer

According to the reviewer’s comment, we included information about Ac2-26 in the Introduction section. In this regard, two new references (#38 and #39) have been now included.

  1. The authors should also better introduce Folfirinox and the authors drugs used in this study. What’s their target and mode of action? How is this connected to Annexin A1? Also is a rather high concentration of 25 µM standard in cell culture experiments?

Authors’ answer

We thank the reviewer for this comment which has allowed us to clarify the choice of FOLFIRINOX treatment and its related experimental procedure. Treatment with FOLFIRINOX (fluorouracil, irinotecan, and oxaliplatin) is generally considered to be a more effective systemic regimen than gemcitabine in pancreatic cancer patients. In separate trials of patients with metastatic pancreatic cancer, FOLFIRINOX therapy had a response rate of 31.6% compared to gemcitabine with a response rate of 23%. Although clinical practice guidelines suggest that either regimen may be delivered as first-line therapy to patients with advanced pancreatic cancer, FOLFIRINOX has been favored in clinical practice, leaving gemcitabine as an alternative for patients who do not tolerate FOLFIRINOX [#34 as new reference]. Numerous research teams have developed in vitro models to decipher pancreatic cancer resistance to gemcitabine [#35 as new reference] while others examined the individual effect of the three drugs included in the FOLFIRINOX protocol [#36; #37 as new references]. These drugs, considered as individually entities, show different impacts on DNA stability by preventing synthesis, causing damage or inhibiting repair. To date, based on literature studies, it is difficult modelling the FOLFIRINOX protocol both in vitro and in vivo and the mechanisms underlying this phenomenon are largely unknown. However, it was previously reported that the FOLFIRINOX effect is immediately correlated to cell cycle alteration with accumulation of the cells in the S phase and decrease of the proportion of cells in G0/G1. In this regard, accumulation of cells in the S and G2/M phases of the cell cycle is a response commonly observed with 5-FU [54]. In our model, by using the concentration of 25 μM FOLFIRINOX in correlation with the 5-FU amount, we have interestingly assessed the spheroids response in absolute tendency with what it has been described above. We have also to underline that, as preliminary investigations, we have analyzed the 2D cell response in time and concentration-curve dependent way finding that at the indicated concentration the major effect has been observed in term of variation of cell cycle and viability (data not shown). Finally, other research teams have used higher FOLFIRINOX concentration, in accordance to the protocol used on patients, in other models of 3D cultures justifying this choice as related to the complex structure of the model with respect to the monolayer [Porter RL, Magnus NKC, Thapar V, Morris R, Szabolcs A, Neyaz A, Kulkarni AS, Tai E, Chougule A, Hillis A, Golczer G, Guo H, Yamada T, Kurokawa T, Yashaswini C, Ligorio M, Vo KD, Nieman L, Liss AS, Deshpande V, Lawrence MS, Maheswaran S, Fernandez-Del Castillo C, Hong TS, Ryan DP, O'Dwyer PJ, Drebin JA, Ferrone CR, Haber DA, Ting DT. Epithelial to mesenchymal plasticity and differential response to therapies in pancreatic ductal adenocarcinoma. Proc Natl Acad Sci U S A. 2019; 116(52):26835–45. doi: 10.1073/pnas.1914915116. Erratum in: Proc Natl Acad Sci U S A. 2020 Jan 13]. The crucial activity of ANXA1 in response to FOLFIRINOX represents for us the appealing outcome which strongly allowed us to consider this protein as a potential prognostic marker to address the therapy protocol. To the best of our acknowledge, no evidence for a possible direct correlation between ANXA1 and FOLFIRINOX exist in pancreatic cancer. All these issues are stressed in the Introduction and Discussion sections to better clarify the major aim of our work. Furthermore, new references have been now included in the revised version of this manuscript (please refer to page 2 of 25 and page 20 of 25).

  1. For cell cycle analysis it may be more intuitive for readers if they label G2 as G2/M as the two phases are hard to tell apart from each other.

Authors’ answer

Following the reviewer suggestion, we modified all the histograms related to the cell cycle including G2/M.

  1. Figure legends (e.g. Figure 3) should be more detailed and should describe which assay was used, treatment details, color keys and stats.

Authors’ answer

Taking advantage of the reviewer’s comment, we enriched the figure legends, particularly for figure 3, with more details about the methodologies which have been used and the kinds of treatments. The colors of the histograms has been changed following further reviewer’s comments by aligning black and grey colors and related nuances.

  1. Instead of using different symbols for statistical comparison between different groups (*, #,@,c, etc.) it would less confusing for readers if the authors stick to one symbol (*) and draw lines above bars connecting samples being compared. Also, did the authors correct stats for multiple testing?

Authors’ answer

Following the reviewer’s comment, we reorganized the significance symbols in each histogram in figure 3, 4, 5, 6. In particular, we replaced the high number of symbols by the use of two kinds of them (* and #), differentiating the related comparisons by different colors and, where possible, by the red or black lines with * symbol. Each change has been also highlighted in the related figure legends. Moreover, we have now included the multiple comparison test we used, in the 2.12 paragraph of the Material and methods section of the revised version of the manuscript. 

  1. Several methods sections are missing including: apoptosis & necrosis assays, STRING analysis

Authors’ answer

We improved the description of apoptosis and necrosis assays and further STRING analysis details were added at the end of 2.5. paragraph of Material and methods section of the revised manuscript.

  1. Line 240: the authors present no data supporting their statement of cellular metabolic activity. Please adjust text accordingly and stick as close as possible to facts supported in figures rather than making unsupported bold statements (several other examples exist throughout the manuscript).

Authors’ answer

Regarding the “cellular metabolic activity” reported at line 240 of the manuscript, we have now improved the text 3.1 paragraph of the Results section (page 6 of 25). Indeed, we used this expression according to the “application note” of IncuCyte Live-Cell Analysis system. We used the CellTiter-Glo 3D kit to perform the ATP end-point viability assay at the end of the experiment. With this assay we have read the ATP relative luminescent units (RLU) values. The luminescence signal increased with cell number and this may reflect changes in cell metabolism upon spheroid formation, particularly once the spheroids become large and develop a hypoxic core [Alcantara SL, Oliver M, Patel K, Dale T, Trezise D, Holtz N, Endsley E. Label-free, real-time live-cell assays for spheroids: IncuCyte® bright-field analysis. APPLICATION NOTE IncuCyte® Live-Cell Analysis System. 2017. Sartorius https://www.sartorius.com/download/876312/8000-0558-a00-spheroids-app-note-ecyslap-1--data.pdf].

Round 2

Reviewer 1 Report

The authors have shown an important effort to improve the manuscript.  The revision indeed clarified almost all the asked questions. However, in my point of view, figure presentation could be improved. Graphs presented at Figure 3 (A and E), Figure 4 (E), Figure 5 (C, D, E and F) and Figure 6 could be re-thought and upgraded. Maybe another type of graph, such as a stacked bar or line, could be a better solution. In addition, information in the X-axis could be denoted as (+) or (-).  The presentation could be cleaner. For instance, Wang et al. (2022) in the Figure 3 - Cell cycle - present an example of data. Sundaramurthi et al. (2022) in the Figure 8 (C and D) presented apoptosis in a stacked graph.

Wang S, Han S, Cheng W, Miao R, Li S, Tian X, Kan Q. Design, Synthesis, and Biological Evaluation of 2-Anilino-4-Triazolpyrimidine Derivatives as CDK4/HDACs Inhibitors. Drug Des Devel Ther. 2022 Apr 11;16:1083-1097. doi: 10.2147/DDDT.S351049. PMID: 35431540; PMCID: PMC9012344.

Sundaramurthi H, García-Mulero S, Tonelotto V, Slater K, Marcone S, Piulats JM, Watson RW, Tobin DJ, Jensen LD, Kennedy BN. Uveal Melanoma Cell Line Proliferation Is Inhibited by Ricolinostat, a Histone Deacetylase Inhibitor. Cancers (Basel). 2022 Feb 3;14(3):782. doi: 10.3390/cancers14030782. PMID: 35159049; PMCID: PMC8833954

Author Response

Reviewer’s comment

The authors have shown an important effort to improve the manuscript.  The revision indeed clarified almost all the asked questions. However, in my point of view, figure presentation could be improved. Graphs presented at Figure 3 (A and E), Figure 4 (E), Figure 5 (C, D, E and F) and Figure 6 could be re-thought and upgraded. Maybe another type of graph, such as a stacked bar or line, could be a better solution. In addition, information in the X-axis could be denoted as (+) or (-).  The presentation could be cleaner. For instance, Wang et al. (2022) in the Figure 3 - Cell cycle - present an example of data. Sundaramurthi et al. (2022) in the Figure 8 (C and D) presented apoptosis in a stacked graph.

Wang S, Han S, Cheng W, Miao R, Li S, Tian X, Kan Q. Design, Synthesis, and Biological Evaluation of 2-Anilino-4-Triazolpyrimidine Derivatives as CDK4/HDACs Inhibitors. Drug Des Devel Ther. 2022 Apr 11;16:1083-1097. doi: 10.2147/DDDT.S351049. PMID: 35431540; PMCID: PMC9012344.

Sundaramurthi H, García-Mulero S, Tonelotto V, Slater K, Marcone S, Piulats JM, Watson RW, Tobin DJ, Jensen LD, Kennedy BN. Uveal Melanoma Cell Line Proliferation Is Inhibited by Ricolinostat, a Histone Deacetylase Inhibitor. Cancers (Basel). 2022 Feb 3;14(3):782. doi: 10.3390/cancers14030782. PMID: 35159049; PMCID: PMC8833954

Authors’ answer

We thank this Reviewer for finding improved the manuscript. Therefore, according to this new comment, we have now modified the indicated figures. In detail, we have changed the graph E in figures 3 and 4 by using the stacked histograms to represent cell cycle. Moreover, we have now modified figure 5 in the graphs C, D, E and F in which we have indicated the experimental points by using the symbols + and -. Then, the whole figure 6 has been adjusted by using stacked graphs in A for WT spheroids (A and B in the previous revised version) and B for ANXA1 KO spheroids (C and D in the previous revised version) in order to define the apoptotic and necrotic cells on the percentage of alive ones. Thus, the following modifications have been included in the related figure legend and in the 3.6 paragraph of the Results section.

We thank again this Reviewer for this helpful comment that allowed us to further improve our manuscript.

Reviewer 2 Report

The reviewers have sufficiently addressed my comments with the exception of the GO-term enrichment analysis suggested in review point 6. Although the authors mention in their rebuttal that they have performed a GO-term enrichment analysis provided in Table S1, they (1) do not cite this analysis in the main text (all data should be discussed or otherwise not included in a manuscript), (2) do not provide a method section in the material and methods section (what were the parameters and source of their analysis) and (3) to this reviewer it seems like the authors simply listed (and color-coded selected categories) all GO-terms of proteins differentially expressed in ANXA1 vs KO proteins. A GO-term analysis should not just list terms associated with target proteins but should include a statistical analysis whether there are certain terms are enriched beyond what would be expected by choice/background. Please provide a separate table where results of a GO-term enrichment analysis for up/downregulated proteins are displayed, mention them in the manuscript (if results interesting consider making a figure) and provide a short description in the methods section.

Author Response

Reviewer’s comment

The reviewers have sufficiently addressed my comments with the exception of the GO-term enrichment analysis suggested in review point 6. Although the authors mention in their rebuttal that they have performed a GO-term enrichment analysis provided in Table S1, they (1) do not cite this analysis in the main text (all data should be discussed or otherwise not included in a manuscript), (2) do not provide a method section in the material and methods section (what were the parameters and source of their analysis) and (3) to this reviewer it seems like the authors simply listed (and color-coded selected categories) all GO-terms of proteins differentially expressed in ANXA1 vs KO proteins. A GO-term analysis should not just list terms associated with target proteins but should include a statistical analysis whether there are certain terms are enriched beyond what would be expected by choice/background. Please provide a separate table where results of a GO-term enrichment analysis for up/downregulated proteins are displayed, mention them in the manuscript (if results interesting consider making a figure) and provide a short description in the methods section.

Authors answer

We thank the reviewer for the observation, and we have accordingly implemented the proteomics results in 3.2 paragraph of the Results section, as well as the methods one (2.5 paragraph of the Material and methods section). Moreover, we have now performed an enrichment analysis via the Database for Annotation, Visualization, and Integrated Discovery (DAVID). The obtained results are now reported in Table S2 and S3 and are discussed in the related paragraphs mentioned above of Results and Material and methods sections.
